# IC-Custom: Diverse Image Customization via In-Context Learning

Yaowei Li[1,4], Xiaoyu Li[2]*, Zhaoyang Zhang[2], Yuxuan Bian[5], Gan Liu[3], Xinyuan Li[3], Jiale Xu[2], Wenbo Hu[2], Yating Liu[6], Lingen Li[5], Jing Cai[3], Yuexian Zou[1,4]*, Yancheng He[3], Ying Shan[2]

[1]ADSP Lab, School of ECE, Peking University    [2]ARC Lab, Tencent PCG    [3]Tencent
[4]Guangdong Provincial Key Laboratory of Ultra High Definition Immersive Media Technology
[5]The Chinese University of Hong Kong    [6]Tsinghua University

Project Page:  https://liyaowei-stu.github.io/project/IC_Custom/

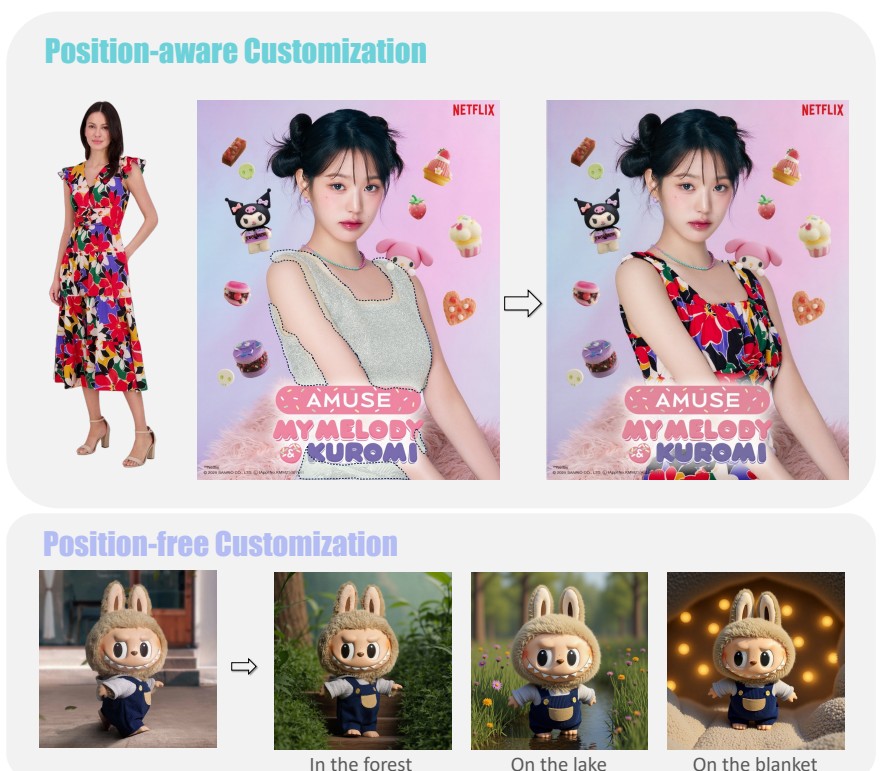

Figure 1: **Visualization of *IC-Custom* results.** Our method supports diverse image customization scenarios, including position-aware (location-specified editing conditioned on a mask) and position-free (ID-consistent generation guided by text) customization.

## Abstract

Image customization, a crucial technique for industrial media production, aims to generate content that is consistent with reference images. However, current approaches conventionally separate image customization into position-aware and position-free customization paradigms and lack a universal framework for diverse customization, limiting their applications across various scenarios. To overcome these limitations, we propose *IC-Custom*, a unified framework that seamlessly integrates position-aware and position-free image customization through in-context learning. *IC-Custom* concatenates reference images with target images to a polyptych, leveraging DiT's multi-modal attention mechanism for fine-grained token-level interactions. We propose the In-context Multi-Modal Attention (ICMA)

---

*Project Lead & Corresponding Authors

mechanism, which employs learnable task-oriented register tokens and boundary-aware positional embeddings to enable the model to effectively handle diverse tasks and distinguish between inputs in polyptych configurations. To address the data gap, we curated a 12K identity-consistent dataset with 8K real-world and 4K high-quality synthetic samples, avoiding the overly glossy, oversaturated look typical of synthetic data. *IC-Custom* supports various industrial applications, including try-on, image insertion, and creative IP customization. Extensive evaluations on our proposed *ProductBench* and the publicly available *DreamBench* demonstrate that *IC-Custom* significantly outperforms community workflows, closed-source models, and state-of-the-art open-source approaches. *IC-Custom* achieves about 72% higher human preference across identity consistency, harmony, and text alignment metrics, while training only 0.4% of the original model parameters.

# 1 INTRODUCTION

Image customization, which ensures that generated content remains consistent with the identity of reference images, has enabled applications such as image insertion (Chen et al., 2024a;b; Mao et al., 2025; Song et al., 2025), IP creation (Ruiz et al., 2023b; Ye et al., 2023b; Tewel et al., 2024; Tan et al., 2024; Mou et al., 2025), and visual try-on (Wang et al., 2024; Guo et al., 2025; Xu et al., 2025). These capabilities are vital for industrial media production, supporting consistent content creation across diverse visual contexts.

Early image customization methods (Ruiz et al., 2023a; Gal et al., 2022; Avrahami et al., 2023) relied on per-instance optimization, which was time-consuming. Subsequent approaches (Ye et al., 2023a; Chen et al., 2024b;a) added control branches to pre-trained diffusion models to inject identity information from reference images. However, these methods were constrained by model architecture and scalability issues, resulting in suboptimal performance. Recently, by leveraging the long-range modeling inductive bias of DiT architectures (Peebles & Xie, 2023b; Esser et al., 2024b; Labs, 2024a), image conditions can be directly input as sequences, interacting with noisy tokens through multi-modal attention mechanisms, without the need for additional branches. This enables image customization methods to exhibit powerful emergent capabilities (Song et al., 2025; Mou et al., 2025; Labs, 2024b;c; Tan et al., 2024; Mao et al., 2025).

Despite these advances, existing methods still face significant challenges in maintaining consistent identity across diverse user requirements and customization scenarios (see Tab. 1): **(1)** They typically treat image customization as two separate tasks. In *position-aware* customization, an reference identity is inserted into masked regions of a fill-in image. In *position-free* customization, identity-consistent images are generated from text prompts. **(2)** They provide limited support for diverse mask types, often confusing user-drawn with precise masks, *e.g.*, treating coarse hand-drawn regions as exact boundaries. These limitations hinder the development of unified frameworks capable of flexibly handling diverse customization requirements, forcing separate models for each scenario and limiting the development of robust, comprehensive identity representations.

Table 1: Comparison of *IC-Custom* with previous image customization methods (Labs, 2024b;c; Tan et al., 2024; Song et al., 2025; Mou et al., 2025; Hurst et al., 2024b). The checkmarks and crosses indicate task compatibility.

| Model | Position-aware | | Position-free |
|---|---|---|---|
| | *precise* | *user-drawn* | |
| FLUX.1 workflow | ✓ | ✓ | ✗ |
| OminiCtrl | ✗ | ✗ | ✓ |
| Insert Anything | ✓ | ✓ | ✗ |
| DreamO | ✗ | ✗ | ✓ |
| GPT-4o | ✗ | ✗ | ✓ |
| *IC-Custom* | ✓ | ✓ | ✓ |

To this end, we propose *IC-Custom*, a unified framework that seamlessly integrates position-aware and position-free image customization, enabling flexible and identity-consistent customization across diverse scenarios (see Fig. 1). Specifically, we first employ a diptych format by concatenating the reference identity image with the fill-in image (either partially or fully masked), yielding a unified representation that allows the model to handle diverse customization settings within a single framework. Building on DiT's multi-modal attention, we further introduce a novel In-Context Multi-Modal Attention (ICMA) module that more effectively transfers identity information from the reference image to the fill-in image and enables comprehensive customization across diverse scenarios. The ICMA module features two key innovations: **(1)** Three types of learnable, task-oriented

register tokens to specify the customization type—position-aware customization (with precise or user-drawn masks) and position-free customization—allowing the model to adapt its behavior based on user requirements. **(2)** Two types of learnable positional embeddings to represent spatial relationships: Reference Embeddings (RE) for the reference identity image and Fill Embeddings (FE) for the fill-in image, helping the model clearly differentiate input boundaries in the diptych format.

To enable effective training of our unified framework, we curated a high-quality dataset *CustomData*, consisting of both real-world and synthetic samples. Specifically, we curated 8K identity-consistent diptychs from real-world sources and an additional 4K synthetic diptychs, resulting in a total of 12K diptychs. This comprehensive dataset enables our model to learn robust identity representations across diverse contexts and viewpoints, while also addressing the limitations of previous methods that overly rely on synthetic data and often produce artificial-looking results.

To extensively evaluate the performance of our method, we use *ProductBench* and *Dream-Bench* (Ruiz et al., 2023a) to assess both position-aware and position-free customization capabilities. *ProductBench* is our manually curated benchmark for position-aware customization, consisting of 40 identity-consistent images with an even distribution of rigid and non-rigid objects, along with their corresponding precise and user-drawn masks. We also use DreamBench to evaluate position-free customization performance. Extensive subjective and objective evaluations demonstrate that *IC-Custom* outperforms community workflows, the closed-source GPT-4o (March 25, 2025), and state-of-the-art open-source approaches. Notably, *IC-Custom* achieves a 72% higher human preference across identity consistency, harmony, and text alignment metrics, while training only 0.3% of the parameters of the pre-trained FLUX model.

In summary, our contributions are as follows:

- We propose a unified framework that seamlessly integrates position-aware and position-free image customization via in-context formulation.
- We introduce the ICMA module, which enables flexible image customization through learnable task-oriented register tokens and boundary-aware positional embeddings.
- We curate a dataset from real-world sources, addressing the limitations of existing methods that rely on synthetic data, which often produce artificial-looking results.
- We demonstrate that our method outperforms existing approaches across a range of metrics, surpassing community workflows, closed-source models, and state-of-the-art open-source methods.

## 2 PRELIMINARIES

**MM-DiT Architecture.** Recent state-of-the-art generative diffusion models, such as SD3 (Esser et al., 2024b) and FLUX (Labs, 2024a), leverage the MM-DiT architecture (Peebles & Xie, 2023a), which integrates a Multi-modal Attention (MMA) mechanism with Rotary Position Embedding (RoPE) as a central component. This design enables the concurrent processing of noisy image tokens $X_t \in \mathbb{R}^{n \times d}$ and text tokens $C_T \in \mathbb{R}^{l \times d}$, as shown in Eq. 1.

$$\text{MMA}\left([X_t; C_T]\right) = \text{softmax}\left(\frac{\mathcal{R}(Q) \cdot \mathcal{R}(K)^\top}{\sqrt{d}}\right) \mathcal{R}(V). \tag{1}$$

Here, $Q$, $K$, and $V$ are derived from the projection of the concatenated input $[X_t; C_T] \in \mathbb{R}^{(n+l) \times d}$, with the operator $\mathcal{R}$ applying RoPE to $Q$ and $K$ to encode positional information.

**Flow Matching.** The model is trained within the Rectified Flow (RF) (Liu et al., 2022). The Continuous Normalizing Flow (CNF) is formalized as the following ODE:

$$\frac{d}{dt}X_t = v(X_t, t)dt = X_1 - X_0, \quad \forall t \in [0, 1]. \tag{2}$$

Here, given a clean latent variable $X_0 \sim p_{\text{data}}$ and a Gaussian noise sample $X_1 \sim \mathcal{N}(0, 1)$, $X_t$ is constructed via linear interpolation:

$$X_t = tX_1 + (1 - t)X_0, \quad \forall t \in [0, 1]. \tag{3}$$

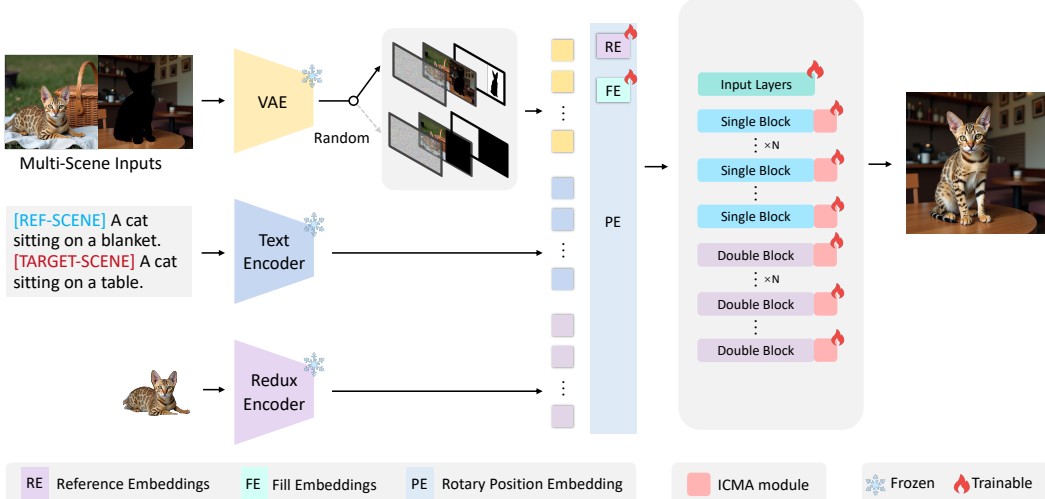

Figure 2: **Model overview.** (1) Our model takes in-context diptych inputs together with redux embeddings and text prompts. (2) During training, it randomly chooses to mask either the entire fill-in image (position-free customization) or only partial regions (position-aware customization) to produce diverse in-context latents. (3) The ICMA module, equipped with task-oriented register tokens and boundary-aware positional embeddings (see Sec. 3.2), is integrated into the architecture. We train LoRA adapters on the ICMA module while unfreezing the input layers.

Subsequently, the Conditional Flow Matching (CFM) loss (Lipman et al., 2023) is employed to train a velocity filed prediction model $v_\Theta$:

$$\mathcal{L}_{\text{CFM}} = \mathbb{E}_{t \sim p(t),\, X_1 \sim \mathcal{N}(0,1),\, (X_0, C_\text{T}) \sim p_{\text{data}}} \left[ \left\| v_\Theta(X_t, C_\text{T}, t) - (X_1 - X_0) \right\|_2^2 \right]. \qquad (4)$$

Here, $t$ is sampled from a *Logit-Normal Distribution* (Esser et al., 2024a) with the probability density function $p(t) = \frac{\exp(-0.5 \cdot (\text{logit}(t) - \mu)^2 / \sigma^2)}{\sigma \sqrt{2\pi} \cdot (1-t) \cdot t}$, where $\text{logit}(t) = \log \frac{t}{1-t}$. From the Logit-Normal Distribution definition, $Y = \text{logit}(t) \sim \mathcal{N}(\mu, \sigma)$, with $\mu = 0$ and $\sigma = 1$ under the RF.

**DiT-based Image Customization Methods** Recent state-of-the-art DiT-based image customization methods (Chen et al., 2024c; Mao et al., 2025; Wu et al., 2025b; Song et al., 2025; Mou et al., 2025), integrate reference image conditions directly into the input via concatenation, instead of using additional network branches. This method unifies reference and other conditions into a single sequence, improving integration during flow matching. However, these methods typically train position-aware and position-free customization tasks separately, without explicitly addressing their potential unification. In position-aware tasks, the identity's location is specified using a mask, while position-free tasks leverage textual guidance to generate identity-consistent content. For instance, ACE++ (Mao et al., 2025) and OmniControl (Tan et al., 2024) train separate LoRA adapters, InsertAnything (Song et al., 2025) is specifically trained for position-aware tasks, and DreamO (Mou et al., 2025) and UNO (Wu et al., 2025b) are designed for position-free tasks.

## 3 METHOD

As shown in Fig. 2, we introduce *IC-Custom*, a novel approach that presents a unified framework for comprehensive image customization, as detailed in Sec. 3.1. At its core, *IC-Custom* leverages In-Context Multi-Modal Attention (ICMA) to effectively adapt to diverse customization scenarios, as described in Sec. 3.2. Additionally, we curate a high-quality dataset for comprehensive customization tasks, sourced from both real-world and synthetic data, with image resolutions exceeding 800×800 pixels, as outlined in Sec. 3.3.

### 3.1 IN-CONTEXT DIPTYCH CUSTOMIZATION

**Motivation.** Formally, position-aware customization can be framed as a reference-guided image filling task, represented as $p(\hat{X} \mid C_\text{I}, C_{\text{I}'}, M)$, where $\hat{X}$ denotes the customized output, $C_\text{I}$ denotes the reference identity image, $C_{\text{I}'}$ represents the image to be filled, and $M$ denotes the mask specifying the filling position. In contrast, position-free customization is viewed as a reference-guided text-to-image task, formalized as $p(\hat{X} \mid C_\text{I}, C_\text{T})$. Since position-free customization can be regarded as a special case of image filling where $M$ and $C_{\text{I}'}$ are set to zero, we unify both paradigms under the formulation $p(\hat{X} \mid C_\text{I}, C_{\text{I}'}, M, C_\text{T})$.

**Diptych Framework and Training Strategy.** Based on the unified formulation above, we introduce an in-context[1] diptych format to unify diverse input conditions and support this paradigm. Specifically, we concatenate the reference identity image $C_\text{I}$ with the fill-in image $C_{\text{I}'}$ in a diptych layout, then encode them jointly as tokens to enforce simultaneous modeling and generation. The model is trained with the following CFM loss:

$$\mathcal{L}_\text{CFM} = \mathbb{E}_{t \sim p(t),\, X_1 \sim \mathcal{N}(0,1),\, (X_0, C_\text{T}) \sim p_\text{data}} \left[ \left\| v_\Theta([X_t, X_0^m, M], C_\text{T}, t) - (X_1 - X_0) \right\|_2^2 \right], \quad (5)$$

where $X_0 = [C_\text{I}; C_{\text{I}'}]$ denotes the width-wise diptych concatenation of the reference identity image and the fill-in image, $X_t$ is computed according to Eq. 3, and $X_0^m = X_0 \odot M$, with $\odot$ indicating element-wise multiplication. Here, $[X_t, X_0^m, M]$ represents the channel-wise concatenation of these three components. The text condition $C_\text{T}$ provides scene descriptions for both the reference identity image and the fill-in image, separated by the placeholders [REF-SCENE] and [TARGET-SCENE]. Notably, during training, instead of requiring triplets $(C_\text{I}, C_{\text{I}'}, \hat{X})$, where $C_{\text{I}'}$ and $\hat{X}$ typically differ in identity, we use two images of the same identity and set $\hat{X} = C_{\text{I}'}$, enabling the model to predict $C_{\text{I}'}$ conditioned on $M$ and $X_0^m$; hence Eq. 5 defines $X_0 = [C_\text{I}; C_{\text{I}'}]$ rather than $X_0 = [C_\text{I}; \hat{X}]$.

Based on this formulation, once paired data $\{C_\text{I}, C_{\text{I}'}, M, C_\text{T}\}$ are available, the model can be trained in two complementary modes without collecting separate datasets or designing distinct model structures. Specifically, setting $C_{\text{I}'}$ and $M$ to zero (i.e., a global mask) corresponds to position-free customization, while using nonzero (localized) masks for $C_{\text{I}'}$ and $M$ enables position-aware customization. Thus, a single paired dataset suffices to support both capabilities through simple variations in training inputs.

### 3.2 IN-CONTEXT MULTI-MODAL ATTENTION

**Challenges.** Although our pipeline seamlessly adapts to diverse customization settings, it still faces several challenges. **(1)** *Task-type ambiguity*: for example, under position-aware customization settings, the model often misinterprets user-drawn masks as precise boundaries, generating content that fully fills and strictly follows the mask shape. **(2)** *Image-boundary confusion*: in diptych prediction settings (Eq. 5), the model struggles to differentiate between reference and target regions, leading to undesirable edge artifacts.

**Proposed ICMA.** To address these issues, we propose In-Context Multi-Modal Attention module (ICMA), a variant of the multi-modal attention mechanism. As illustrated in Fig. 3 (a), ICMA incorporates two key design innovations: **(1)** *learnable task-oriented register tokens* to explicitly indicate the customization type (precise masks, user-drawn masks, or position-free); and **(2)** *learnable boundary-aware positional embeddings*—comprising Reference Embeddings (RE) and Fill Embeddings (FE)—to encode spatial relationships between the reference identity image and the fill-in image. Formally, the ICMA mechanism operates as follows:

$$\begin{aligned}
\mathcal{P}(x) &= x + [\mathcal{E}_\text{R}; \mathcal{E}_\text{F}] + \mathcal{R}(x), \\
Q &= [\mathcal{P}(Q_\text{I}); \; Q_\text{T} + \mathcal{R}(Q_\text{T})], \\
K &= [\mathcal{P}(K_\text{I}); \; K_\text{T} + \mathcal{R}(K_\text{T})]; \; \mathbf{r}_i], \\
V &= [V_\text{I}; \; V_\text{T}; \; \mathbf{r}_i], \\
h' &= \text{MHA}(Q, K, V),
\end{aligned} \quad (6)$$

---

[1] Here, "in-context" refers to concatenating images and jointly conditioning on their captions.

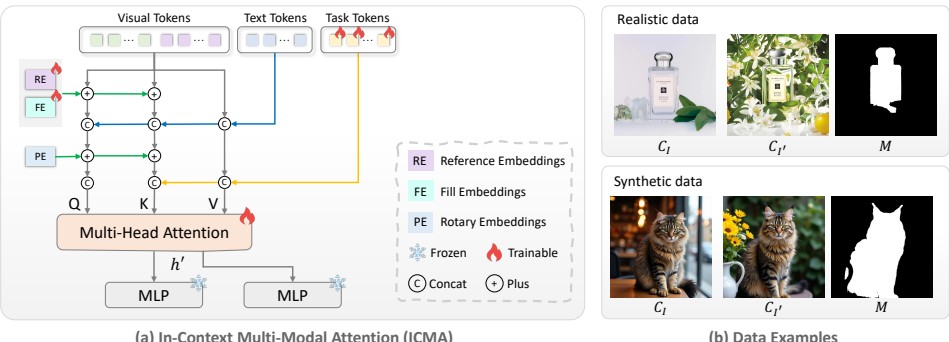

**(a) In-Context Multi-Modal Attention (ICMA)**          **(b) Data Examples**

Figure 3: **(a) In-Context Multi-Modal Attention (ICMA).** ICMA incorporates learnable task-oriented register tokens and boundary-aware positional embeddings (RE, FE) into the multi-modal attention of MM-DiT (Peebles & Xie, 2023a) to specify customization types and delineate input boundaries. **(b) Training data examples.** High-quality identity-consistent quadruples $\{C_{\mathrm{I}}, C_{\mathrm{I'}}, M, C_{\mathrm{T}}\}$ from real-world and synthetic data; for clarity, text descriptions $C_{\mathrm{T}}$ are omitted.

where $[;]$ denotes diptych concatenation, $\mathcal{R}(\cdot)$ denotes rotary position encoding (Su et al., 2024); $Q_I, K_I, V_I \in \mathbb{R}^{n \times d}$ and $Q_T, K_T, V_T \in \mathbb{R}^{l \times d}$ are the query, key, and value matrices for image and text tokens, respectively; $\mathcal{E}_{\mathrm{R}}, \mathcal{E}_{\mathrm{F}}$ are the learnable Reference and Fill embeddings; $\mathbf{r}_i \in \mathbb{R}^{m \times d}$ denotes the $i$-th learnable task-oriented register token; and $\mathrm{MHA}(\cdot)$ is the Multi-Head Attention operation. Our proposed ICMA module replaces the multi-modal attention layers in both the double-block and single-block components of the original FLUX.1 MM-DiT architecture (Labs, 2024a).

## 3.3 IN-CONTEXT CUSTOMIZATION DATA CURATION

**Data Collection.** The scarcity of high-quality customization data remains a critical bottleneck in developing robust customization models. Existing approaches (Tan et al., 2024; Wu et al., 2025b; Li et al., 2025) rely predominantly on synthetic data for training; however, such data often struggles to preserve identity consistency and photorealistic quality, thereby limiting model effectiveness.

To address this challenge, we introduce *CustomData*, a high-quality customization dataset designed for both authenticity and diversity. We curate nearly 8K identity-consistent realistic image pairs from e-commerce platforms, covering real-world scenarios such as clothing try-on, cosmetics, furniture, electronics, accessories, home decor, and personal care products, with resolutions ranging from $800 \times 800$ to $3000 \times 3664$ pixels. To further enrich the dataset and extend coverage beyond commercial products, we add 4K high-quality, identity-consistent synthetic pairs carefully filtered from the SynCD $1024 \times 1024$ subset (Kumari et al., 2025), resulting in a comprehensive dataset of 12K $\{C_{\mathrm{I}}, C_{\mathrm{I'}}, M, C_{\mathrm{T}}\}$ samples (see Fig. 3(b) for visualization; symbol definitions in Sec. 3.1).

**Data Processing.** Our filtering process applies three rules: (1) exclude items whose DI-NOv2 (Oquab et al., 2023) feature similarity between $C_{\mathrm{I}}$ and $C_{\mathrm{I'}}$ is below 0.2; (2) discard pairs composed entirely of blank-background images; and (3) ensure $C_{\mathrm{I'}}$ is not a blank-background image. These rules improve identity consistency and reduce ambiguity. We then use Qwen-VL2.5 (Bai et al., 2025) to auto-generate captions for *CustomData* (system prompt in Appendix Sec. J) and Grounded SAM (Ren et al., 2024) to obtain ground-truth masks, while randomly generating user masks under predefined rules to support model training (see Appendix Sec. L for details).

## 4 EXPERIMENTS

### 4.1 EXPERIMENTS SETUP

**Implementation Details.** *IC-Custom* builds on the pre-trained text-to-image model FLUX.1-Fill (Labs, 2024b). We train LoRA (Hu et al., 2022) (rank 64) on the first 10 layers of both single and double blocks, while directly fine-tuning the image and text input layers. In total, only 49.26M parameters are trainable—just $0.4\%$ of the original FLUX model's 12B parameters (19 double and 38

Table 2: **Quantitative results on position-aware and position-free image customization.** Evaluation on *ProductBench* (precise/user-drawn masks) and *DreamBench* shows that *IC-Custom* consistently outperforms existing methods across all objective metrics (higher is better ↑). Baselines: FLUX.1 workflow (Labs, 2024b;c), OminiCtrl/DreamO/Insert Anything (Tan et al., 2024; Mou et al., 2025; Song et al., 2025), GPT-4o (Hurst et al., 2024b).

| Method | ProductBench | | | | | | DreamBench | | |
|---|---|---|---|---|---|---|---|---|---|
| | *Precise Mask* | | | *User-drawn Mask* | | | *Position-free* | | |
| | DINO-I ↑ | CLIP-I ↑ | CLIP-T ↑ | DINO-I ↑ | CLIP-I ↑ | CLIP-T ↑ | DINO ↑ | CLIP ↑ | CLIP-T ↑ |
| FLUX.1 workflow | 60.80 | 81.66 | 31.13 | 62.26 | 81.60 | 31.29 | — | — | — |
| OminiCtrl | 57.93 | 76.06 | 31.31 | — | — | — | 48.29 | 75.85 | 36.82 |
| DreamO | 62.98 | 78.86 | 31.25 | — | — | — | 57.69 | 76.33 | 36.24 |
| Insert Anything | 62.71 | 81.65 | 31.24 | 61.21 | 81.75 | 31.44 | — | — | — |
| GPT-4o | 61.40 | 78.53 | 30.72 | 62.05 | 79.87 | 30.58 | 54.31 | 77.38 | 36.33 |
| **IC-Custom (Ours)** | **63.14** | **81.92** | **31.75** | **63.28** | **81.95** | **31.80** | **65.67** | **83.19** | **36.88** |

Table 3: (a) **Human-study results on image customization quality (higher is better)**. (b) **Ablation studies on ProductBench.** Abbreviations: Zero-shot = zero-shot inference without fine-tuning; w/o IL = without training Input Layers; w/o RD = without using Real Data for training; w/o UM = without using User-drawn Mask for training; w/o TR = without Task-oriented Register tokens; w/o PE = without Boundary-aware Positional Embeddings.

**(a) Human-study results**

| Method | Consistency ↑ | Harmony ↑ | Text Alignment ↑ |
|---|---|---|---|
| FLUX.1 workflow | 3.2% | 5.3% | — |
| OminiCtrl | 1.5% | 2.1% | 6.3% |
| DreamO | 5.4% | 3.2% | 10.1% |
| Insert Anything | 6.8% | 6.5% | — |
| GPT-4o | 4.6% | 7.5% | 21.4% |
| **IC-Custom (Ours)** | **78.5%** | **75.4%** | **62.2%** |

**(b) Ablation on ProductBench**

| Models | Precise Mask | | | User-drawn Mask | | |
|---|---|---|---|---|---|---|
| | DINO-I ↑ | CLIP-I ↑ | CLIP-T ↑ | DINO-I ↑ | CLIP-I ↑ | CLIP-T ↑ |
| Zero-shot | 55.49 | 77.55 | 31.24 | 57.63 | 79.84 | 31.20 |
| w/o IL | 62.00 | 81.52 | 31.36 | 62.13 | 81.33 | 31.64 |
| w/o RD | 62.38 | 81.81 | 31.62 | 62.71 | 81.85 | 31.22 |
| w/o UM | 62.65 | 81.82 | 31.58 | 61.30 | 81.28 | 31.64 |
| w/o TR | 63.00 | 81.42 | 31.43 | 63.07 | 81.44 | 31.33 |
| w/o PE | 62.99 | 81.31 | 31.42 | 63.08 | 81.40 | 31.30 |
| **Ours** | **63.14** | **81.92** | **31.75** | **63.28** | **81.95** | **31.80** |

single blocks). Unlike prior methods (Song et al., 2025; Mou et al., 2025) that train LoRA on all layers (e.g., DreamO (Mou et al., 2025) trained 707M parameters), our approach drastically cuts training cost. The model is optimized on our 12K dataset for 20K iterations using AdamW (Loshchilov & Hutter, 2017) with a learning rate of $5 \times 10^{-5}$ and a batch size of 4. To handle diverse resolutions, we employ a data-bucketing strategy that groups samples by size (e.g., 800×800, 1024×1024, 1024×1280, 1280×1280, 1504×1504) so each batch has uniform input dimensions. We also present a web application and inference pipeline in Appendix M.

**Benchmarks.** To assess our model's performance in both position-aware and position-free customization settings, we evaluate on our proposed *ProductBench* and the open-source *DreamBench* (Ruiz et al., 2023a) benchmark. *ProductBench* contains 40 high-quality, identity-consistent items with resolutions exceeding $1024 \times 1024$ pixels. Each item includes paired images and corresponding masks, with no overlap with our training data. We use SAM (Kirillov et al., 2023) to annotate precise masks and manually create user-drawn masks. The dataset is evenly divided into rigid and non-rigid categories, covering diverse domains such as clothing try-on, accessories, bags, furniture, toys, and perfume, specifically designed to evaluate position-aware customization. *DreamBench* comprises 30 items, each with 5–6 identity-consistent images and used to evaluate position-free customization. We take the first image of each item as the reference. Additionally, we use Qwen-VL2.5 (Bai et al., 2025) to generate in-context textual descriptions for both benchmarks. For *ProductBench*, we directly prompt it to caption the diptych input, whereas for *DreamBench* we prompt it to creatively generate new scene descriptions. (see Appendix Sec. K for details)

**Metrics.** Follow established methods (Ruiz et al., 2023a; Wu et al., 2025b), we consider 3 objective evaluation metrics across two aspects: identity consistency, and text alignment.

- Identity Consistency: We calculate the DINO-I Score (Oquab et al., 2023) and CLIP-I (Radford et al., 2021) Score between reference images and generated images to assess identity preservation.
- Text Alignment: We use the CLIP-T score (Radford et al., 2021) to evaluate the model's instruction-following ability.

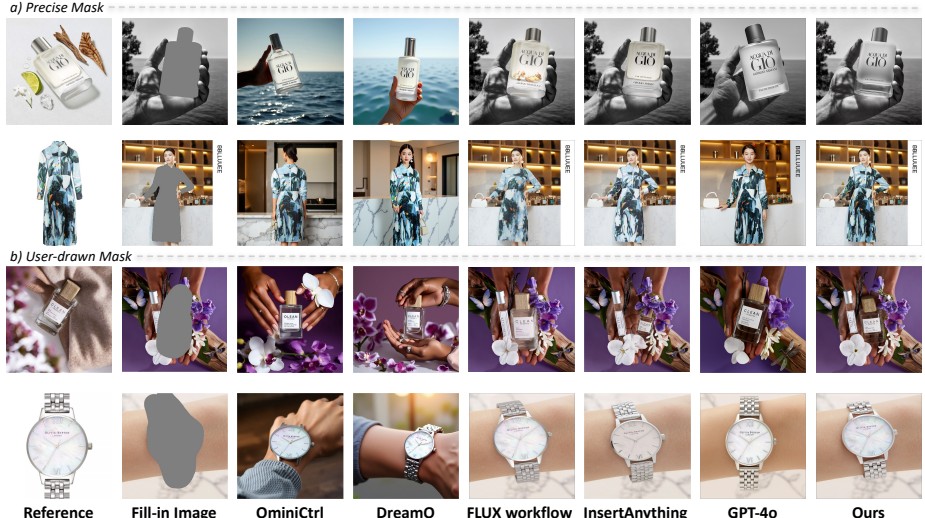

Figure 4: **Qualitative comparison of position-aware customization under precise-mask and user-drawn-mask settings.** OminiCtrl and DreamO lack support for fill-in inputs. *IC-Custom* achieves high-quality customization with harmonious lighting, shadows, and perspectives.

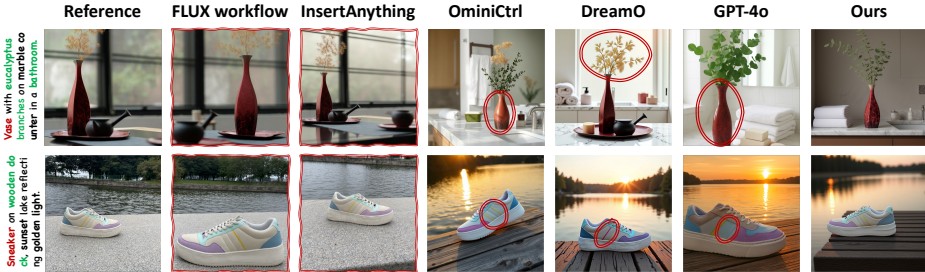

Figure 5: **Qualitative comparison on position-free customization.** *IC-Custom* achieves more realistic, coherent, and detailed customization. Red circles highlight incorrect regions or details.

We also incorporate subjective evaluation metrics: identity consistency, harmony, and text alignment to assess the performance of customization models.

**Baselines.** We compare our approach against several strong baselines, including the community FLUX.1 workflow (FLUX.1-Fill with FLUX.1-Redux) (Labs, 2024b;c), state-of-the-art DiT-based open-source methods OminiCtrl (Tan et al., 2024), DreamO (Mou et al., 2025), and Insert Anything (Song et al., 2025), as well as the commercial system GPT-4o (Hurst et al., 2024a) (March 25, 2025). Among them, FLUX.1 workflow and Insert Anything are primarily designed for position-aware customization, whereas OminiCtrl and DreamO target position-free customization. Beyond evaluating each method in its native setting, we also adapt the other baselines to complementary scenarios—feeding blank fill-in images to FLUX.1 workflow and Insert Anything to approximate position-free customization, and prompting OminiCtrl and DreamO with text descriptions of the identity embedded in the fill-in image scene to approximate position-aware customization. GPT-4o, in contrast, is a unified vision–language system. We therefore provide it with alternating image–text pairs and explicit instructions to perform each customization mode.

## 4.2 POSITION-AWARE CUSTOMIZATION

**Quantitative Comparisons.** Tab. 2 reports quantitative results on *ProductBench* using both precise and user-drawn masks. *IC-Custom* achieves state-of-the-art identity consistency and text alignment, particularly under the more practical user-drawn mask setting (e.g., DINO-I 63.28 vs. 62.26).

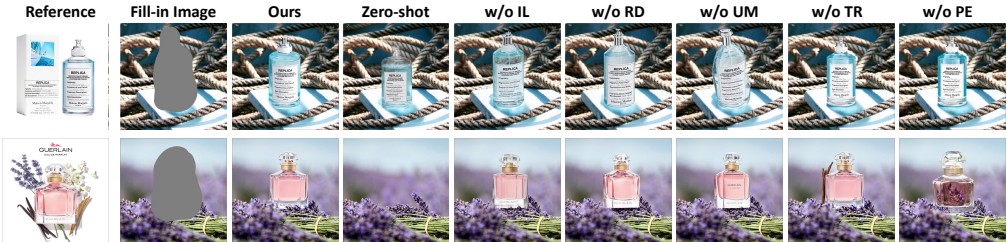

Figure 6: **Ablation Visualization.** Qualitative results show that our model preserves identity consistency while enabling harmonious customization. Abbreviations are as follows: Zero-shot = zero-shot inference without fine-tuning; w/o IL = without training Input Layers; w/o RD = without using Real Data for training; w/o UM = without using User-drawn Mask for training; w/o TR = without Task-oriented Register tokens; w/o PE = without Boundary-aware Positional Embeddings.

Although the adapted OminiCtrl, DreamO, and GPT-4o achieve reasonable scores, they essentially regenerate images rather than perform reference-based image filling (see the following paragraph). Despite being specifically designed for position-aware customization, FLUX.1 workflow and Insert Anything still underperform compared with our method.

**Qualitative Comparisons.** Fig. 4 presents qualitative comparisons on *ProductBench*. OminiCtrl, DreamO, and GPT-4o tend to regenerate entire images rather than perform position-aware customization, for example in the precise-mask try-on case (second row) where the human's face is completely altered. FLUX.1 workflow and Insert Anything also produce noticeable artifacts and weaker identity preservation compared with our model. Moreover, under the user-drawn mask setting, our method generates content with harmonious size, shape, and appearance instead of merely filling the mask region. Thanks to its unified in-context formulation, *IC-Custom* delivers position-aware customization with harmonious lighting, shadows, textures, and materials. More visual results are provided in Appendix Sec. H.

### 4.3 POSITION-FREE CUSTOMIZATION

**Quantitative Comparisons.** FLUX.1 workflow and Insert Anything lack position-free customization capability and, even after adaptation, merely replicate the reference (see the following paragraph), so we exclude them. As shown in the DreamBench section of Tab. 2, OminiCtrl shows poor identity consistency (low DINO-I and CLIP-I), while DreamO and GPT-4o, though strong, still lag behind our approach. Trained on a high-quality mix of real and synthetic data with a unified customization representation, our method achieves state-of-the-art performance across all metrics.

**Qualitative Comparisons.** Figure 5 presents qualitative comparisons in the position-free setting. FLUX.1 workflow and Insert Anything fail to achieve true position-free customization, tending instead to replicate the reference identity image. OminiCtrl and DreamO produce results that are less realistic and less coherent than ours, while GPT-4o, despite strong instruction-following capabilities, sometimes loses fine-grained identity details. In contrast, *IC-Custom* consistently generates diverse, harmonious, and identity-consistent results. More visual results are provided in Appendix Sec. H.

### 4.4 HUMAN EVALUATION

We conducted a user study with 20 participants on 50 randomly selected samples from both position-aware and position-free subsets. For each sample, participants were asked to identify the best-performing model across three dimensions: identity consistency, harmony, and text alignment. As shown in Tab. 3(a), our method receives the highest human preference across all three dimensions compared with existing approaches. As FLUX.1 workflow and Insert Anything only take images as input, we exclude them from the rating of text alignment.

Figure 7: **Effect of Boundary-aware Positional Embeddings.** Without Boundary-aware Positional Embeddings (PE), position-free customization can produce blurred or ambiguous boundaries between the reference and generated content. Incorporating these embeddings sharpens boundaries.

## 4.5 ABLATION STUDIES

We present ablation studies of *IC-Custom* in Tab. 3(b), examining model architecture, training data sources, and training strategies. We first establish zero-shot performance as a baseline. We then validate several key design choices: ❶ Without training the DiT image and text input layers (w/o IL), the model struggles to transfer the pre-trained diffusion prior to customization tasks, especially under user-drawn mask settings; ❷ Training solely on synthetic data (w/o RD) weakens identity consistency and realism; ❸ Omitting user-drawn mask data during training (w/o UM) substantially reduces performance on free-form masks; ❹ Removing Task-oriented Register tokens (w/o TR) or Boundary-aware Positional Embeddings (w/o PE) also degrades performance.

Qualitative results in Fig. 6 confirm these findings: all ablated variants introduce artifacts or shape distortions, whereas our full model preserves identity while naturally integrating it into the scene, yielding harmonious lighting and perspective. While the ablation variants exhibit different imperfections, a representative case is w/o TR: Fig. 6 (second row) shows unwanted structures forming around the imprecise user-drawn mask boundaries, where background completion is expected. Adding TR helps alleviate this by enhancing the model's ability to distinguish and adapt to different task types. We also observe that, in position-free customization, performing flow matching on both the reference and output images tends to blur their boundaries—an issue alleviated by incorporating Boundary-aware Positional Embeddings (see Fig. 7). We also include ablation studies under different input-conditioning settings in Appendix Sec. B. IC-Custom remains robust and reliable across all input-conditioning modes.

## 5 DISCUSSION

This paper presents *IC-Custom*, a flexible and effective framework for image customization. Our approach introduces four key contributions: (1) an in-context customization paradigm that unifies position-free and position-aware image customization; (2) a novel In-Context Multi-Modal Attention (ICMA) mechanism to adapt to different customization settings; (3) a high-quality identity-consistent dataset sourced primarily from real-world images; and (4) an evaluation benchmark with a balanced distribution of rigid and non-rigid customization tasks. Extensive experiments demonstrate that *IC-Custom* achieves state-of-the-art performance across multiple metrics.

Despite these achievements, our method does not explicitly model viewpoint, lighting, geometry, or other 3D scene properties, which we plan to address in future work. We also provide an initial exploration of multi-reference customization in Appendix F. In addition, Appendix E presents a preliminary study on geometric consistency, Appendix G illustrates the cross-domain customization capability of our model, and Appendix D demonstrates its multi-round iterative customization performance. We also include representative failure cases and corresponding analyses in Appendix I.

## ACKNOWLEDGMENTS

This work is supported by NSFC (No. 62176008), Tencent University Relations (Tencent AI Lab RBFR2024006) and Guangdong Provincial Key Laboratory of Ultra High Definition Immersive Media Technology (Grant No. 2024B1212010006).

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

# Appendix

## CONTENTS

## A  RELATED WORK

### A.1  DIFFUSION MODELS

Recent advances in diffusion models (Sohl-Dickstein et al., 2015; Ho et al., 2020) have set new benchmarks in generative modeling, surpassing traditional approaches such as Variational Autoencoders (VAE) (Kingma et al., 2013) and Generative Adversarial Networks (GANs) (Goodfellow et al., 2020). As a result, many state-of-the-art generative methods (Dhariwal & Nichol, 2021; Ho et al., 2020; Nichol et al., 2021; Ramesh et al., 2022) now rely on diffusion models as their core framework. Early approaches utilized a U-Net (Ronneberger et al., 2015) architecture with cross-attention, achieving competitive performance and efficiency. The open-sourcing of Stable Diffusion (Rombach et al., 2022) has significantly accelerated research in this area. More recently, diffusion models have been further advanced through the integration of transformer architectures, as demonstrated by models like SD3 (Esser et al., 2024b), FLUX (Labs, 2024a), CogVideo (Yang et al., 2024b) and WAN (Wan et al., 2025). These diffusion-transformer hybrid models (Vaswani et al., 2017) have shown even greater performance and are now widely applied to a range of tasks, including depth estimation (Bochkovskii et al., 2024; Yang et al., 2024a; Hu et al., 2025), editing (Labs et al., 2025; Wu et al., 2025a), compositing (Labs, 2024b; Yang et al., 2025) and more.

### A.2  IMAGE CUSTOMIZATION

Image customization is typically accomplished by integrating additional control signals from reference images into text-to-image foundation models. One line of work (Wu et al., 2025b; Li et al., 2025; Hurst et al., 2024a; Mou et al., 2025; Tan et al., 2024; Chen et al., 2024c) focuses on position-free customization, directly generating identity-consistent images based on input reference images and text, as seen in GPT-4o (Hurst et al., 2024a), DreamO (Mou et al., 2025), and OminiControl (Tan et al., 2024). However, these methods struggle with position-aware customization, particularly when a masked source image is provided, as they cannot preserve the unedited regions. In contrast, methods like Insert Anything (Song et al., 2025) and the FLUX.1-Fill-Redux workflow (Labs, 2024b) specialize in position-aware customization, inserting subjects into masked source images, but lack the capability for position-free customization. Concurrent works such as ACE++ (Mao et al., 2025) and FLUX.1 Kontext (Labs et al., 2025) share similar ideas with our approach, yet differ in innovative technical details. In this work, we propose a flexible framework that can address both position-aware customization and position-free customization. We also propose a data curation pipeline to collect high-quality real image data from different product images. Benefiting from this framework and high-quality data, our model achieves highly identity consistent customization, which can be used in real production.

## B  ABLATION ON DISTINCT INPUT CONDITIONING MODES

Our method supports not only the in-context diptych reference but also the integration of both the Redux Encoder and the Text Encoder. To assess how IC-Custom behaves under different combinations of textual and reference-based inputs, we conduct a comprehensive ablation across **four input-conditioning configurations** within the position-aware customization setting. ❶ **Full conditions** (default setting), ❷ **Redux-only** (no text), ❸ **Text-only** (no Redux reference), ❹ **Diptych-only** (neither text nor Redux; relies solely on the in-context diptych reference). Table 4 summarizes the quantitative comparison on ProductBench under both *precise masks* and *user-drawn masks*. Across all four input modes, IC-Custom achieves **stable and reliable customization performance**. Even the diptych-only configuration yields competitive results, indicating that the in-context diptych framework already provides strong customization capability, while the Redux and text inputs act as optional boosters. As shown in Fig. 8, the qualitative comparisons further validate that our method supports all four input-conditioning modes: the full-conditions setting is the most stable overall, yet the reduced-input variants also perform strongly.

Table 4: **Comparison of IC-Custom under four input-conditioning modes in the position-aware setting.** Across all configurations, the model exhibits consistent and robust performance.

| Setting | Precise Mask | | | User-drawn Mask | | |
|---|---|---|---|---|---|---|
| | DINO-I ↑ | CLIP-I ↑ | CLIP-T ↑ | DINO-I ↑ | CLIP-I ↑ | CLIP-T ↑ |
| **Full conditions (default)** | 63.14 | **81.92** | **31.75** | 63.28 | **81.95** | **31.80** |
| Redux-only | **63.34** | 81.49 | 31.37 | **63.78** | 81.86 | 31.10 |
| Text-only | 62.96 | 81.32 | 31.42 | 63.04 | 81.43 | 31.29 |
| Diptych-only | 62.97 | 81.41 | 31.41 | 63.18 | 81.77 | 31.30 |

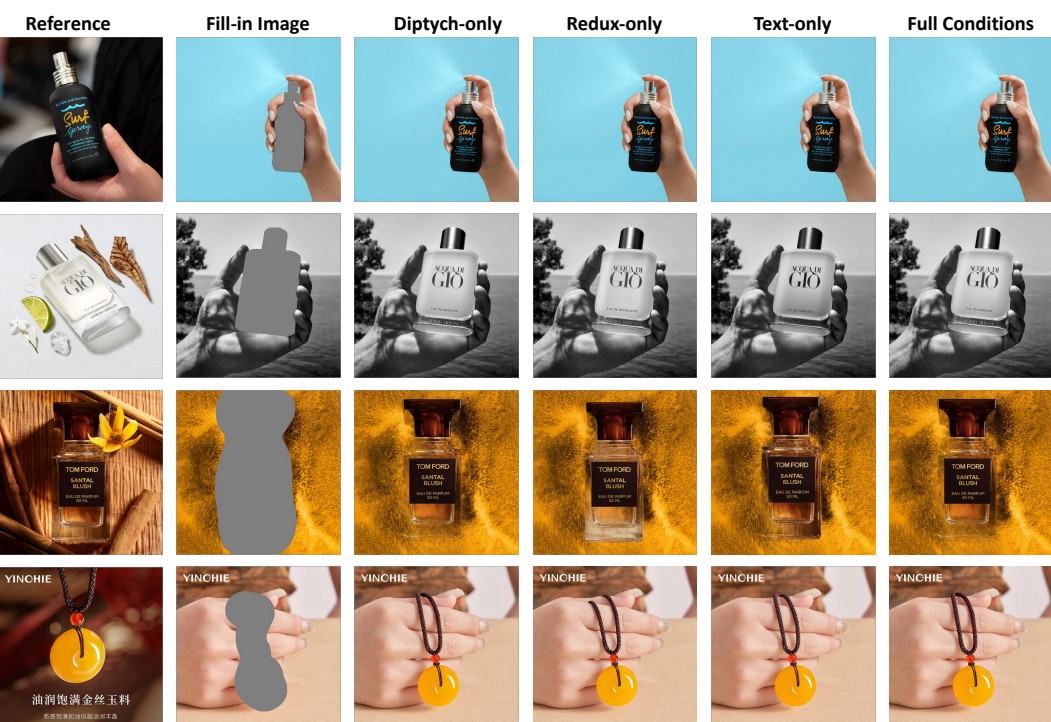

Figure 8: **Qualitative comparison of IC-Custom under four input-conditioning modes in the position-aware setting.** Full conditions yield the most stable results, while the reduced-input modes (Redux-only, text-only, diptych-only) remain strong and coherent.

## C  COMPARISON WITH ACE++

ACE++ (Mao et al., 2025) is a concurrent work proposing the Long-context Condition Unit (LCU), conceptually similar to our in-context diptych. However, ACE++ focuses on four separate domain-specific tasks and trains distinct LoRA adapters for each, rather than a unified model handling both position-aware and position-free customization. Moreover, unlike our framework, ACE++ does not incorporate the innovative ICMA module. For a fair comparison on *ProductBench*, we directly use ACE++'s publicly released **subject LoRA adapters** to evaluate its performance under our benchmark. As shown in Tab. 5 and Fig. 9, our model consistently produces more identity-consistent and visually coherent customization results, showing superior perspective, lighting, and shape fidelity while operating as a single unified model rather than multiple task-specific LoRA adapters.

## D  MULTI-ROUND ITERATIVE CUSTOMIZATION

In the position-aware customization setting, our framework is inherently *composable*, allowing multi-round, iterative refinement of the generated content. As illustrated in Fig. 10, IC-Custom

Table 5: **Comparison with ACE++ (Mao et al., 2025) on ProductBench.** Metrics under Precise Mask (left) and User-drawn Mask (right); higher is better (↑).

| Precise Mask | | | | User-drawn Mask | | | |
|---|---|---|---|---|---|---|---|
| **Method** | **DINO-I** | **CLIP-I** | **CLIP-T** | **Method** | **DINO-I** | **CLIP-I** | **CLIP-T** |
| ACE++ | 60.68 | 81.34 | 31.64 | ACE++ | 61.26 | 81.16 | 31.42 |
| **Ours** | **63.14** | **81.92** | **31.75** | **Ours** | **63.28** | **81.95** | **31.80** |

Figure 9: **Qualitative comparison with ACE++ (Mao et al., 2025). Our method produces more identity-consistent and harmonious customization results.** We compare our unified framework with ACE++ on *ProductBench*.

can perform GPT-style, multi-object customization by sequentially applying the model in multiple rounds. We begin by generating an initial customization result based on the in-context diptych reference. This intermediate output can then be reused as a new reference input in subsequent rounds, enabling the model to introduce additional objects or refine fine-grained appearance details step-by-step. By iteratively composing these rounds, users can achieve complex, multi-object customization scenarios while preserving spatial coherence and identity consistency throughout the process.

## E GEOMETRIC CONSISTENCY ASSESSMENT VIA 3D RECONSTRUCTION

To explicitly validate whether the customized results generated by our method maintain geometric consistency with the reference identity, we conduct a 3D reconstruction analysis using VGGT (Wang et al., 2025). Specifically, we feed both the reference and generated images into VGGT to assess the recoverability of the underlying 3D structure. As illustrated in Figure 11, VGGT effectively aggregates geometric cues from the generated distinct views, successfully reconstructing 3D point clouds with faithful shapes. Notably, despite the absence of explicit 3D modeling, these results indicate that our method not only preserves visual semantics but also achieves robust geometric alignment with the reference subject. Nevertheless, we acknowledge that our current framework does not support explicit control over the pose of the customized results. Such precise viewpoint controllability pertains to the domain of generative rendering, which lies outside the scope of this paper and is reserved for future work.

## F PRELIMINARY STUDY ON MULTI-REFERENCE CUSTOMIZATION

Benefiting from the learnable task-oriented register tokens and boundary-aware positional embeddings introduced in our **In-Context Multi-Modal Attention (ICMA)**, our model can accurately distinguish customization types and the boundaries between inputs and outputs. This naturally extends to **multi-reference customization**, where multiple reference images of the same identity (but from different scenes) are provided—not as multi-image fusion, but as separate context cues. By aggregating information from multiple references, our model better preserves identity fidelity and

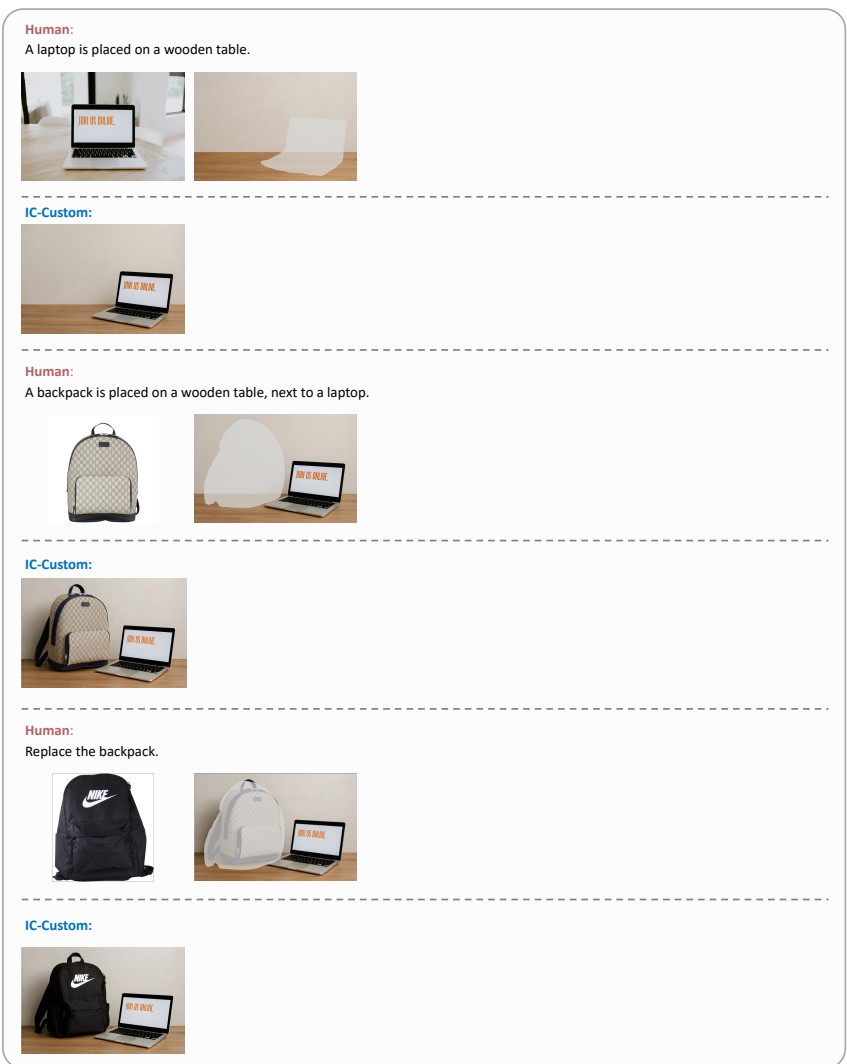

Figure 10: Illustration of multi-round iterative customization: add laptop → add backpack → replace backpack, demonstrating IC-Custom's extended application for multi-object customization.

fine details. To support this setting, we concatenate multiple reference images with the fill-in noise input and introduce an additional **index embedding** in the boundary-aware positional embeddings to differentiate reference indices. Formally, we extend the diptych structure to polyptych, modifying Eq. 5 to $X_0 = [C_I; C_{I'_1}, C_{I'_2}, \ldots, C_{I'_n}]$. Additionally, to distinguish different references, we draw inspiration from DreamO (Mou et al., 2025) by introducing learnable Index embeddings. The key difference is that we add these embeddings in the ICMA module rather than the input tokens. Specifically, we extend the function $\mathcal{P}(x)$ in Eq. 6 as $\mathcal{P}(x) = x + [\mathcal{E}_R + \mathcal{E}_I; \mathcal{E}_F] + \mathcal{R}(x)$, where $\mathcal{E}_I \in \mathbb{R}^{k, m/k \times d}$ are the learnable Index embeddings, $\mathcal{E}_R \in \mathbb{R}^{m \times d}$ and $\mathcal{E}_F \in \mathbb{R}^{n \times d}$ are the learnable Reference and Fill embeddings. We also curated a multi-reference dataset containing **2K real-world** and **2K synthetic polyptychs** for training. In Fig. 12, our multi-reference approach aggregates information from multiple references (e.g., different viewpoints) to better preserve object identity details and textures. This preliminary exploration highlights the broader capability and scalability of our unified customization model, and we plan to further explore this direction in future work.

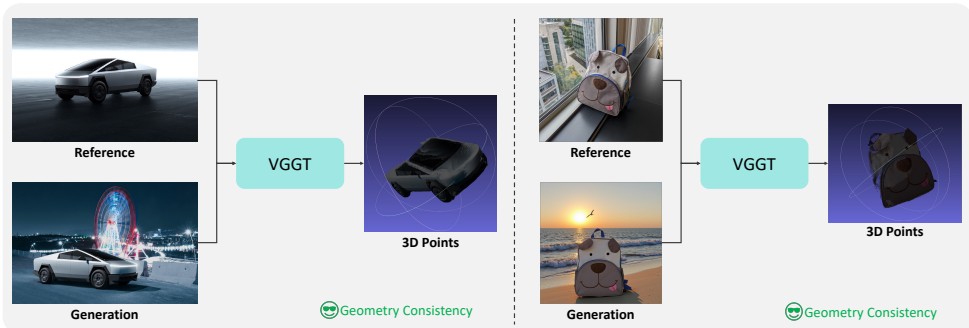

Figure 11: **Evaluation of Geometric Consistency.** Visual comparison of 3D reconstructions derived from reference and generated images.

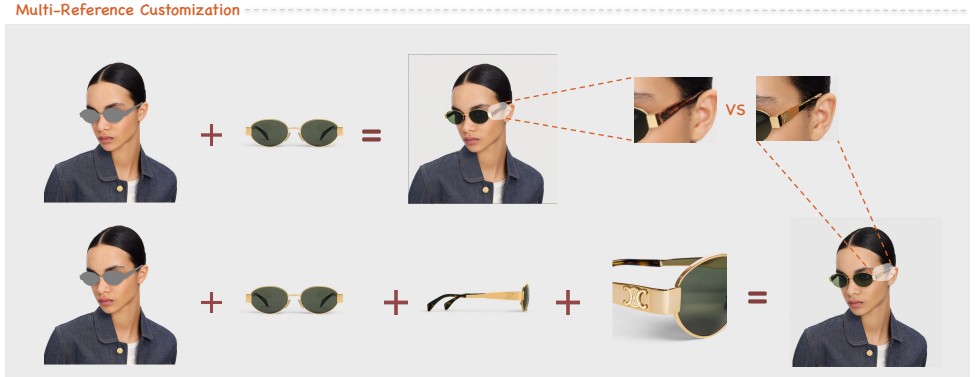

Figure 12: **Multi-Reference Customization.** By aggregating multiple reference images of the same identity from different environments and viewpoints, our model preserves richer details and textures. For example, when a single reference view omits the glasses' temples, the model must hallucinate them; with multiple viewpoints including the temples, it reconstructs the object more completely.

## G   CROSS-DOMAIN CUSTOMIZATION CAPABILITY

To further validate the Cross-Domain Customization Capability of our method, we conducted experiments on position-aware customization tasks across multiple domains, including cross-style, cross-pose, and cross-object scenarios. Fig. 13 highlight the robustness and flexibility of our method in handling a wide range of cross-domain customization tasks. However, we clarify that explicit style-content disentanglement is not a primary objective of our work. As such, it is challenging to specify a reference style image as input to achieve more specific style-driven customization.

## H   ADDITIONAL VISUALIZATION RESULTS

Figure 17 shows additional position-free customization results, where our model seamlessly generates novel scenes that preserve the reference identity based on text descriptions. Figure 18 presents additional position-aware customization results, demonstrating its ability to accurately insert or edit images with different materials and textures while maintaining identity consistency.

## I   FAILURE CASES

While our method demonstrates robust performance across diverse scenarios, we acknowledge specific limitations. We present several failure cases where the performance of the model degrades under certain conditions.

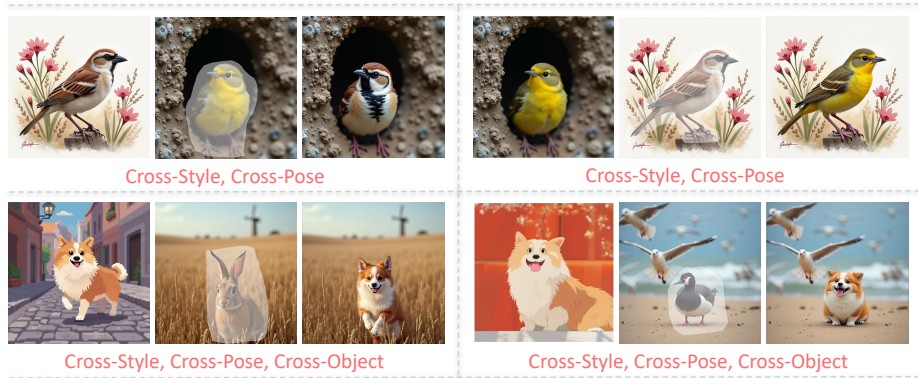

Figure 13: **Qualitative results of cross-domain customization.** Demonstrating the effectiveness of our method in handling cross-style, cross-pose, and cross-object customization tasks.

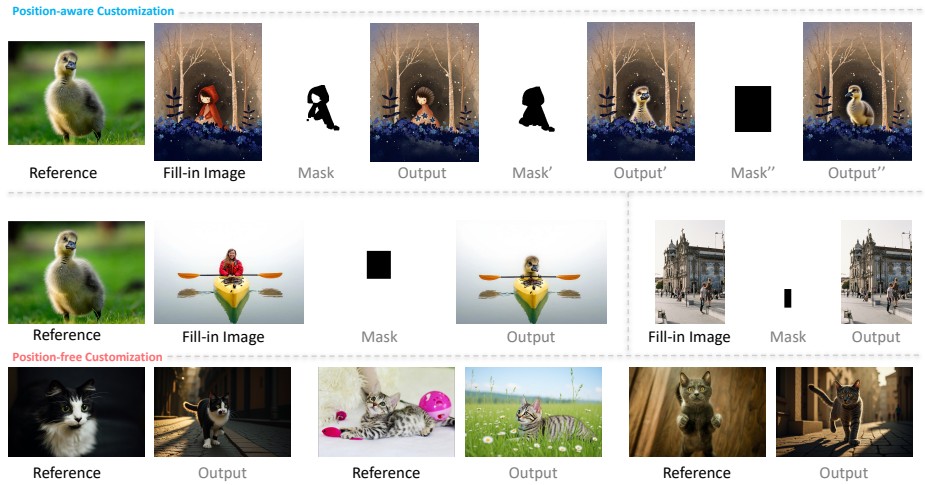

Figure 14: **Failure cases.** We present several failure cases that highlight the limitations of our method and suggest areas for potential improvement.

As illustrated in Fig. 14, in **position-aware customization**, challenges arise when the user-provided mask is **ambiguous, extremely incomplete, or excessively small**. For instance, in the first row of Fig. 14, an ambiguous or incomplete mask leads to the generation of erroneous content instead of the intended duckling; however, dilating the mask into a rectangular bounding box significantly improves performance. Furthermore, as shown in the second column of the second row, when the mask is excessively small, the model fails to inject the target subject entirely due to insufficient position guidance.

In **position-free customization**, the model's performance is hindered when the reference identity lacks **essential details** or features **complex textures**. For example, in the position-free customization row of Fig. 14, the texture patterns of the generated cat are not perfectly consistent with the reference image, indicating a difficulty in faithfully reconstructing high-frequency details for complex subjects. We aim to further investigate these boundary conditions and enhance the robustness of our framework in future work.

**Human**:

I've shown you 2 images that contains the same object. Create a short description of this 2-scene image. Each scene detail should be marked with [SCENE-1], [SCENE-2] indicating the start of each scene's description. Don't use phrases like 'the first image' or 'the second image' as the images are unordered - describe each independently without referencing their order. Focus on describing the key visual contents, effects, colors, textures and styling details of each object. The descriptions for [SCENE-1], [SCENE-2] should be completely independent and should not reference each other. The description should not exceed 512 words. Please strictly follow this format without any additional content.

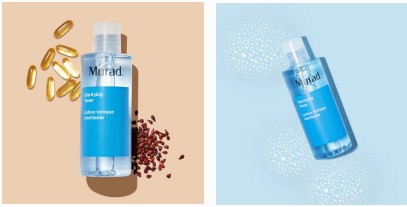

**VLM (Qwen-VL)**:

[SCENE-1] The Murad Clarifying Toner bottle is set against a neutral beige background. The bottle is upright, and the label faces forward, clearly displaying the product information. The golden capsules are scattered around the bottle, adding a touch of contrast to the warm-toned background. The lighting is soft and even, highlighting the transparency of the bottle and the vibrant blue of the label.

[SCENE-2] The Murad Clarifying Toner bottle is placed on a light blue background, creating a cool-toned setting. The bottle is also upright, and the label is visible, showing the product name and description. The background features a subtle pattern of water droplets, enhancing the fresh and hydrating theme associated with the toner. The overall ambiance is serene and refreshing, complementing the product's purpose.

Figure 15: **Example of automated text prompt annotation.** A concatenated pair of identity-consistent images is fed into Qwen-VL2.5 (Bai et al., 2025) with custom-designed instructions to generate corresponding captions for our data.

## J AUTOMATED CAPTIONING FOR DATA

We use Qwen-VL2.5 (Bai et al., 2025) to automatically generate text annotations for our data. Specifically, each concatenated pair of identity-consistent images is fed into Qwen-VL2.5 with custom-designed instructions to generate captions, as illustrated in Fig. 15.

## K AUTOMATED CAPTIONING FOR BENCHMARK

Our benchmark consists of two parts: *ProductBench* for evaluating position-aware customization and DreamBench (Ruiz et al., 2023a) for evaluating position-free customization. For *ProductBench*, we apply the captioning approach described in Sec. J to generate input captions. For DreamBench, which targets position-free customization, we provide the reference image together with prompts designed to elicit creative yet identity-consistent outputs; an example of this prompting strategy is shown in Fig. 16.

## L TRAINING STRATEGY: MASK SAMPLING AND AUGMENTATION

To enhance model flexibility and robustness, we randomly sample mask types during training: position-aware masks with a probability of 0.6 and position-free masks with 0.4. Within the position-aware cases, we further draw user-drawn masks with 0.75 probability and precise masks with 0.25, assigning higher probabilities to harder tasks to provide more training iterations. In addition, we convert precise masks from Grounded SAM into user-drawn masks via standard image-morphology operations such as dilation, erosion, opening, and closing.

## M WEB APPLICATION

We implement a web application using Hugging Face Gradio [2] to provide a simple and seamless interface for both position-free and position-aware customization (see Fig. 19 and Fig. 20). Users

---

[2] https://www.gradio.app/

Figure 16: **Example of DreamBench captioning and generated output.** We illustrate our prompting process for DreamBench, where a reference image and custom instructions are provided to a vision–language model to generate creative, identity-consistent captions. The figure also shows an example image generated by our method using the curated reference and caption.

first select a customization mode and upload a reference image. In the **position-aware mode**, they choose a mask type (precise or user-drawn), upload the fill-in image, optionally refine the mask (via SAM for precise masks or manual brushing for user-drawn masks), and provide an optional text prompt before running the model. In the **position-free mode**, users directly supply a text prompt describing the desired scene or use the built-in VLM-based prompt auto-generation tool prior to execution. This web application provides a simple, unified interface for both position-aware and position-free customization, enabling users to interactively explore our model's capabilities with minimal setup. We will release the full code and the web application as open source to support reproducibility and community adoption.

## N    Ethics Statement

This work complies with the ICLR Code of Ethics.[3] Our study does not involve human or animal subjects, personally identifiable information, or sensitive demographic attributes. All datasets are either publicly available or internally curated, and will be verified for proper licensing prior to open-sourcing. We also adopt the SafeChecker from the Diffusers FLUX.1 framework to filter potentially harmful outputs (e.g., sexual, violent, or toxic content) and apply similar precautions during data collection to minimize such content. We adhere to established research integrity practices, including reproducibility, transparency, and proper attribution of prior work.

## O    Reproducibility Statement

To ensure reproducibility, we provide detailed descriptions of our data preparation and processing in Sec. 3.3, and implementation details in Sec. 4.1, including training hyperparameters, evaluation

---

[3]https://iclr.cc/public/CodeOfEthics

protocols, and baselines clarification. In Appendix Sec. J, we also describe the prompts used when preparing data with the multi-modal language model. We will release our code and models under appropriate licenses to facilitate full reproducibility.

## P  LLM USAGE STATEMENT

In preparing this paper, we used large language models (LLMs), including ChatGPT (Hurst et al., 2024a) and Gemini (Team et al., 2023), solely as writing-assistance tools. Specifically, we first drafted the content ourselves and then used LLMs with prompts such as "You are an expert in academic writing. Please help me refine and rephrase the text to make it more professional, fluent, clear, and readable." We then manually reviewed and revised all LLM outputs to ensure that the text accurately reflects our intended meaning. No part of the research design, experiments, analysis, or results was generated by LLMs; their use was limited to improving clarity and readability of the manuscript. We, the authors, take full responsibility for the content of this paper.

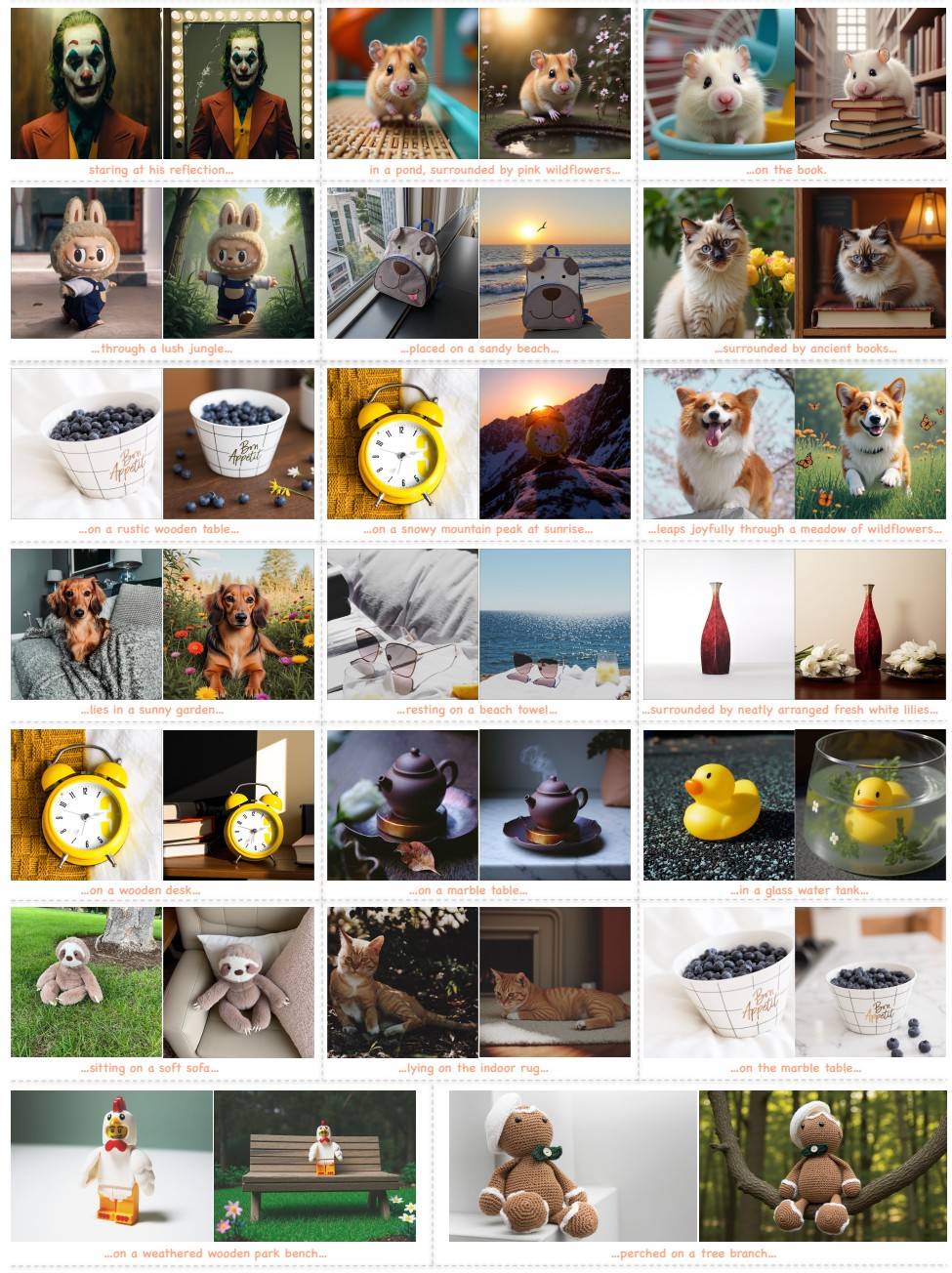

Figure 17: **Additional visualization results on position-free customization.** Our method successfully maintains identity consistency while generating diverse scenes and poses.

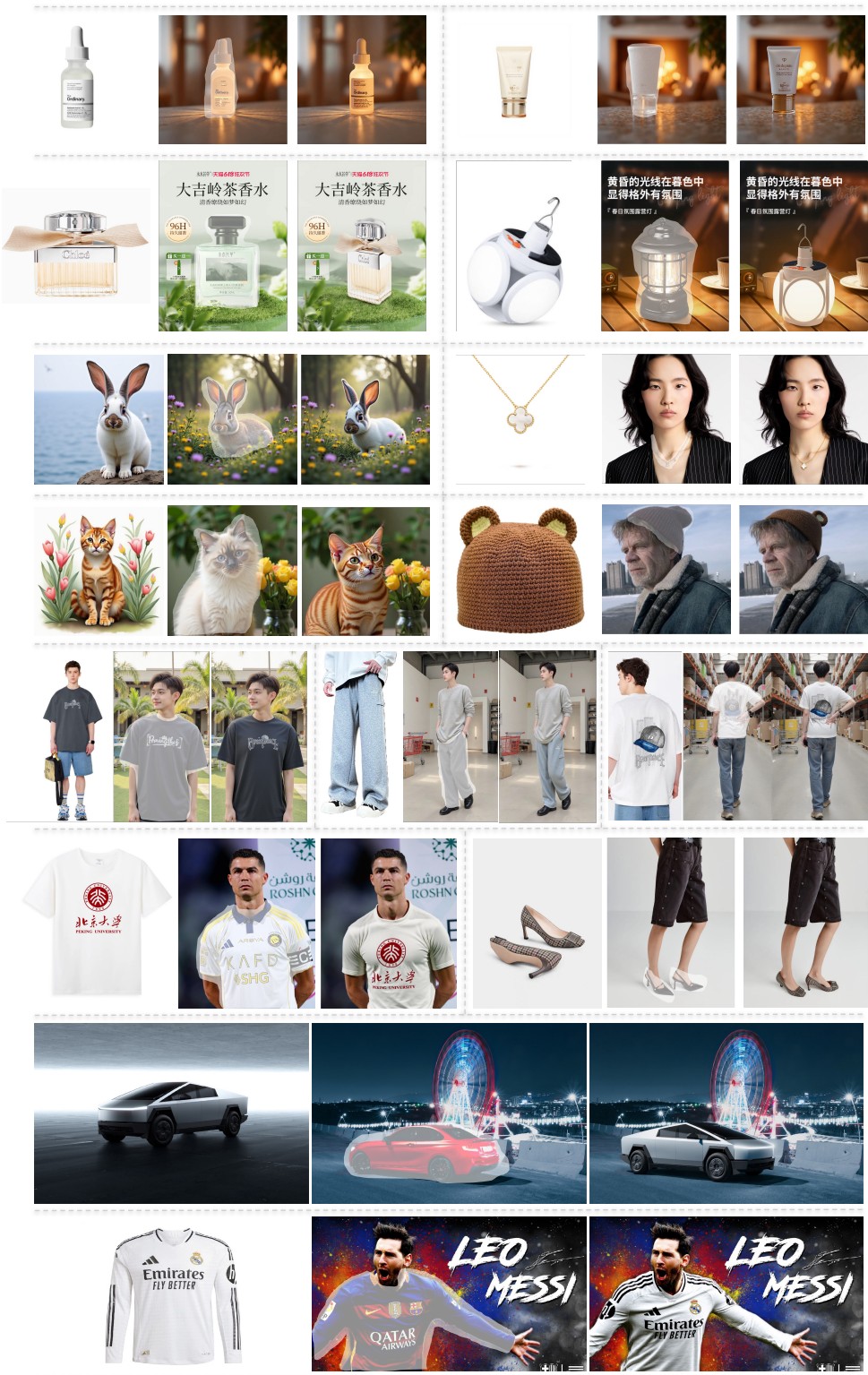

Figure 18: **Additional visualization results on position-aware customization.** Our method successfully maintains identity consistency while seamlessly integrating subjects into diverse lighting, styles, and poses in target scenes.

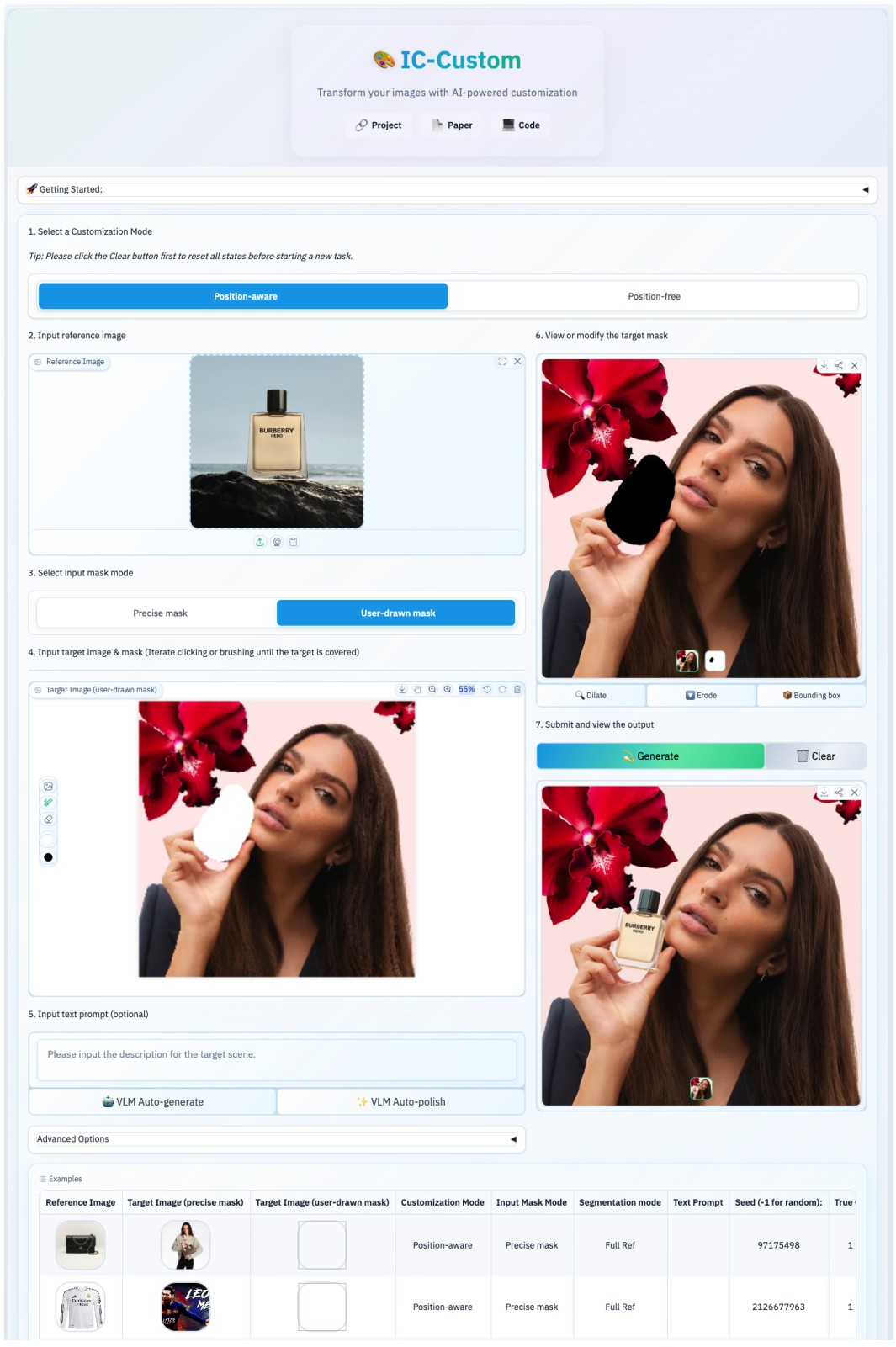

Figure 19: **Web App – Position-aware mode.** Users upload a reference image and a fill-in image, choose the mask type (precise or user-drawn), optionally edit or refine the mask, add an optional text prompt, and then run the model to perform position-aware customization.

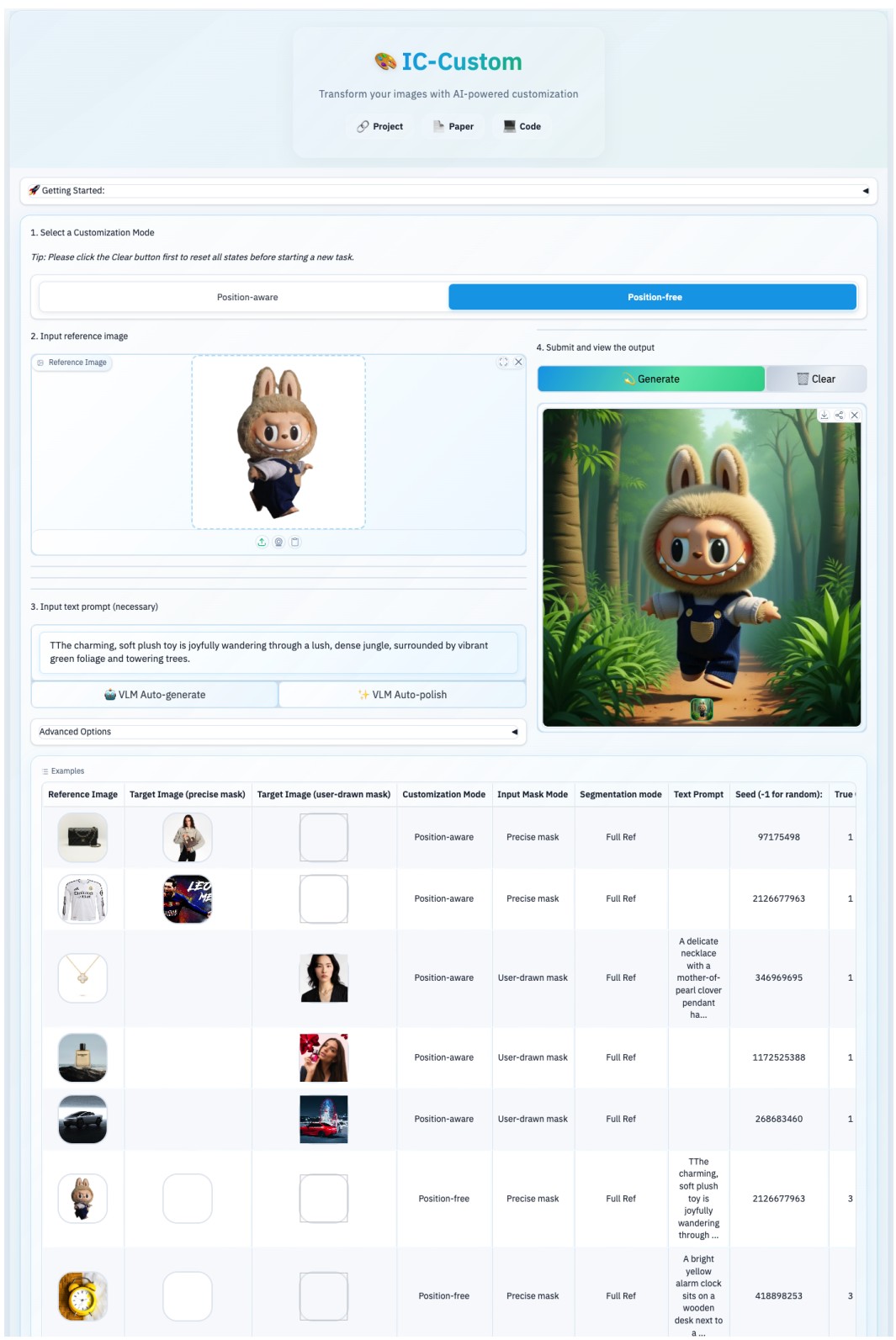

Figure 20: **Web App – Position-free mode.** Users upload a reference image, provide a text prompt describing the desired scene or use the built-in VLM prompt generator, and then run the model to perform position-free customization.

