# OpenReview forum: "IC-Custom: Diverse Image Customization via In-Context Learning"
_ICLR.cc/2026/Conference — ICLR 2026 Poster_

### Official Review · Reviewer_pHSt · 2025-10-30

**Soundness:** 3
**Presentation:** 3
**Contribution:** 3
**Rating:** 6
**Confidence:** 4

**Summary:**

This paper presents IC-Custom, a unified framework for both position-aware and position-free customized generation. The approach formalizes the task under a universal equation, $p(\hat{X}|C_{I},C_{I'},M,C_{T})$, and utilizes an In-Context Diptych Format where the reference identity image and the fill-in image are concatenated and jointly encoded into token sequences using the DiT architecture. The key technical contribution is the In-Context Multi-Modal Attention (ICMA) module, which incorporates learnable task-oriented register tokens to explicitly signal the customization type and Boundary-aware Positional Embeddings to help delineate the input regions in the diptych configuration.

**Strengths:**

1. The paper proposes a unified framework that seamlessly integrates position-aware and position-free image customization, addressing the key limitation of non-unified, task-specific approaches found in prior work.
2. The diptych framework and its associated training strategy are well-designed and appear reasonable for achieving the model's unified capability.
3. The motivation for using both task-oriented register tokens and boundary-aware positional embeddings within the architecture is clear.

**Weaknesses:**

1. The paper lacks sufficient qualitative ablation studies to clearly validate the effect of the Task-oriented Register tokens (TR) and Boundary-aware Positional Embeddings (PE). While quantitative ablation results are provided in Table 3(b), the observed metric values are not significantly compelling compared to the full model design. More visualizations for the ablation studies should be provided to demonstrate the necessity and efficacy of these modules.
2. The Redux Encoder is integrated into the framework; it remains unclear whether the resulting performance in identity consistency is highly reliant on the embeddings provided by this Redux Encoder.

**Questions:**

See Weaknesses

---

> ### Author Response · Authors · 2025-11-20
>
> We thank the reviewer for this highly valuable suggestion.
>
> > **Q1: [Weakness 1]** The paper lacks sufficient qualitative ablation studies to clearly validate the effect of the Task-oriented Register tokens (TR) and Boundary-aware Positional Embeddings (PE). While quantitative ablation results are provided in Table 3(b), the observed metric values are not significantly compelling compared to the full model design. More visualizations for the ablation studies should be provided to demonstrate the necessity and efficacy of these modules.
>
> **A1:**
>
> **[Clarification]:**
> We fully agree that these visualizations are important for understanding the necessity and effectiveness of TR and PE. In addition to the quantitative ablation results reported in **Tab. 3(b)**, the initial submission already included corresponding **qualitative ablation analyses** in the appendix, covering the core components **Task-oriented Register tokens (TR)** and **Boundary-aware Positional Embeddings (PE)**.
>
> **[Revisions]:**
> Since ICLR allows extending the main paper to 10 pages during the discussion phase, we have **moved these qualitative ablations into the main paper (Sec. 4.5, Fig.6 and Fig.7)** and **added additional explanations and visual results**, all marked in **red** for clarity.
>
> **[Findings]:**
> - **w/o TR** introduces noticeable artifacts. For instance, in position-aware customization with user-provided masks, the model should place a plausible object within the masked region and complete the surrounding area as background. However, w/o TR, we observe unwanted structures emerging around the imprecise mask boundaries, indicating reduced task awareness (Fig.6 second row).
> - **w/o PE** may lead to **ambiguous or inconsistent boundaries** (Fig.7), and in some cases degrades the fidelity of the customized object.
>
>
>
> > **Q2: [Weakness 2]** The Redux Encoder is integrated into the framework; it remains unclear whether the resulting performance in identity consistency is highly reliant on the embeddings provided by this Redux Encoder.
>
> **A2:**
>
> **[Clarification]:**
> The Redux Encoder is *not* the primary source of identity preservation in our framework. In fact, one of our main baselines—**the FLUX.1 workflow (FLUX.1-Fill + FLUX.1-Redux)**—is a widely used community pipeline, and its results are reported in **Tab. 2, Tab. 3, Fig. 4, and Fig. 5**. As these results show, **simply incorporating the Redux Encoder is insufficient** to maintain strong identity consistency, nor can it support **both position-aware and position-free customization** in a unified manner.
>
> **[Ablation on Redux Dependency]:**
>  To directly evaluate the influence of Redux on IC-Custom, we added a **Redux-free ablation** in the revised manuscript (**Appendix Sec. B, Tab. 4, Fig. 8**). The results demonstrate that **our method retains strong customization capability even without the Redux Encoder**.
>
>
> **[Key Components Driving Performance]:**
> These findings confirm that the performance gains of IC-Custom primarily come from: 1) the **in-context diptych paradigm**,  2) the **proposed ICMA module**, and 3) **high-quality training data and tailored training strategies**, rather than reliance on the Redux Encoder.

---

### Official Review · Reviewer_JiRo · 2025-10-30

**Soundness:** 3
**Presentation:** 3
**Contribution:** 3
**Rating:** 6
**Confidence:** 3

**Summary:**

The paper introduces IC-Custom, a unified framework for integrating position-aware (mask-guided insertion) and position-free (text-guided generation) image customization tasks using in-context learning on Flux1. architectures. It proposes an In-Context Multi-Modal Attention (ICMA) module with learnable task-oriented register tokens and boundary-aware positional embeddings to handle diverse scenarios. The authors curate a 12K identity-consistent dataset and evaluate on a new ProductBench for position-aware tasks and DreamBench for position-free, demonstrating superior performance over baselines OmniCtrl, DreamO, Insert Anything, and GPT-4o in metrics such as DINO-I, CLIP-I/T, and human preferences.

**Strengths:**

1. Novel unification of two customization paradigms using in-context learning on DiT, with innovative ICMA incorporating task-specific register tokens and positional embeddings to handle ambiguity.

2. Robust dataset curation (12K samples, real+synthetic) and comprehensive evaluations, including ablations that validate key components.

3. Intuitive diptych formulation and clear training strategy enable seamless handling of diverse tasks without separate models.

4. Advances efficient, identity-consistent generation for practical applications, outperforming SOTA with minimal parameters trained.

**Weaknesses:**

1. No quantitative analysis of inference time, failure cases beyond visuals. The paper does not address how long the model takes to generate results, nor does it provide a comparison to baseline models in terms of speed.

2. Unclear Handling of Multi-Reference Customization. The paper briefly mentions multi-reference customization as future work, but does not provide sufficient detail on how this will be incorporated into the current framework.

3. Limited Exploration of Dataset Diversity. The paper mentions that the authors curated a dataset with 12K identity-consistent samples, combining real-world and synthetic data. However, there is limited discussion about the diversity of this dataset, particularly with respect to edge cases.

**Questions:**

1. Could the author share more specific details on the computational resources used, such as the type and number of GPUs, as well as the total GPU hours required for the 20K iterations?
2. In the ablation studies, why do the results show only a minor drop when real data is excluded? How was the synthetic data filtered to ensure it matches the quality of real-world data?
3. The metrics from the user study are significantly higher than those of the baselines; could the author clarify why IC-Custom outperforms them to such a large extent?
4. How does IC-Custom perform when faced with more challenging or extreme customization scenarios, such as very complex images, unclear user input, or ambiguous mask regions?

---

> ### Author Response · Authors · 2025-11-20
>
> We thank the reviewer for raising this meaningful and important question.
>
> > **Q1: [Weakness 1]** The paper does not address how long the model takes to generate results, nor does it provide a comparison to baseline models in terms of speed.
>
> **A1:**
>
> **[Inference Time]:**
> Our method takes approximately **30 seconds** for inference on a single **NVIDIA H20 GPU** with a resolution of **576x1024px** and **25 timesteps**. For comparison, under the same settings, the inference times for the following models are as follows:
> - **DreamO**: 30 seconds
> - **InsertAnything**: 31 seconds
> - **OminiCtrl**: 33 seconds
>
> **[Analysis]:**
> At equivalent settings (without using quantized models), the inference times for these methods are nearly identical. This is because our primary pre-trained model, as well as the models from the **FLUX** series, share similar architecture and model size, leading to comparable performance in terms of speed.
>
>
> > **Q2: [Weakness 2]** Unclear Handling of Multi-Reference Customization. The paper briefly mentions multi-reference customization as future work, but does not provide sufficient detail on how this will be incorporated into the current framework.
>
> **A2:**
>
> **[Revision & Details]:**
> To provide clearer insights into the multi-reference customization approach, we have now expanded this section in the revised manuscript, providing more details on the multi-reference customization preliminary study, including the mathematical formulation of the approach. The updated content can be found in **Appendix Sec.F and Fig.12**. We invite the reviewer to refer to these sections for further details.
>
> **[Clarification]:**
> We presented multi-reference customization as a **potential extension** of our framework, providing preliminary experiments in the appendix to illustrate the model's potential and to inspire future exploration in this direction.
>
> > **Q3: [Weakness 3]** Limited Exploration of Dataset Diversity. The paper mentions that the authors curated a dataset with 12K identity-consistent samples, combining real-world and synthetic data. However, there is limited discussion about the diversity of this dataset, particularly with respect to edge cases.
>
> **A3:**
>
> **[Dataset Diversity]:**
> Our 12K identity-consistent samples include both **real-world** and **synthetic** data. The **real-world data** consists of high-resolution images of common product types such as bags, perfumes, clothes, and rigid objects. The **synthetic data** comes primarily from **SynCD**, which includes 75,000 rigid category assets from **Objaverse** and 16 deformable super-categories of animals, with approximately 100 different subspecies.  Despite filtering, we have retained a large number of diverse categories, especially animals, which are challenging to obtain in real-world datasets.
>
> While the real-world data ensures consistent identity information, the synthetic dataset provides a broader range of categories, including more edge cases. This combination allows us to cover both consistency in identity and diversity in category representation.

---

> ### Author Response · Authors · 2025-11-20
>
> > **Q4: [Questions 1]** Could the author share more specific details on the computational resources used, such as the type and number of GPUs, as well as the total GPU hours required for the 20K iterations?
>
> **A4:**
>
> For training on high-resolution data (above 800px), we used a single machine with **8 NVIDIA H20 GPUs**, and a total batch size of **32**. The total GPU hours required for the 20K iterations were approximately **160 hours**.
>
>
> > **Q5: [Questions 2]** In the ablation studies, why do the results show only a minor drop when real data is excluded? How was the synthetic data filtered to ensure it matches the quality of real-world data?
>
> **A5:**
>
> **[Qualitative Ablation]:**
> **DINO-Score** and **CLIP-Score** are not ideal for **fine-grained evaluation** of identity consistency, as they may miss subtle identity differences, particularly with very similar objects. To address this, we have included **ablation visualizations** (now in **Sec. 4.5, Fig. 6** in the revised manuscript). These visualizations demonstrate that without real data, the generated results can become overly sharp and fail to blend naturally with the environment, underscoring the importance of real data for preserving identity consistency.
>
> **[Synthetic Data Quality]:**
> The **SynCD dataset** itself undergoes filtering to enhance the quality of synthetic data. We performed an additional round of filtering to further ensure **identity consistency** and **image quality** (Sec.3.3). This additional filtering helps ensure that the synthetic data not only supplements data diversity but also maintains strong identity consistency, avoiding any degradation in image coherence.
>
>
> > **Q6: [Questions 3]** The metrics from the user study are significantly higher than those of the baselines; could the author clarify why IC-Custom outperforms them to such a large extent?
>
> **A6:**
>
> **[Findings]:**
> As demonstrated in **Fig. 4 and Fig. 5**, IC-Custom significantly outperforms baselines across both tasks:
> * **Position-Aware Customization:** **OminiCtrl** and **DreamO** lack support for this task. **FLUX workflow** and **InsertAnything** suffer from visible artifacts, while **GPT-4o** tends to alter non-identity context.
> * **Position-Free Customization:** **FLUX** and **InsertAnything** are inapplicable, whereas **OminiCtrl**, **DreamO**, and **GPT-4o** exhibit noticeable identity drift.
>
> In contrast, our approach consistently yields harmonious and identity-preserving results in both scenarios.
>
> **[Analysis]:**
> **DINO-I** and **CLIP-I** lack sensitivity for fine-grained intra-category identity, often failing to reflect significant visual improvements. Our method's superior performance stems from the **ICMA module** and **high-quality training data**, ensuring robust customization that is best appreciated through visual inspection rather than these coarse metrics.
>
>
> > **Q7: [Questions 4]** How does IC-Custom perform when faced with more challenging or extreme customization scenarios, such as very complex images, unclear user input, or ambiguous mask regions?
>
> **A7:**
>
> **[New Validation]:**
> In the revised manuscript, we have added comprehensive evaluations of challenging scenarios:
> * **Appendix Fig. 13** demonstrates performance on complex, cross-domain inputs.
> *  **Appendix Fig. 18** showcases difficult tasks such as virtual try-on (e.g., necklaces, shoes) using coarse, user-drawn masks.
>
> These examples demonstrate how IC-Custom performs in more challenging and extreme scenarios, showing its robustness.
>
> **[Limitations & Failure Cases]:**
> However, our method has some limitations in certain extreme cases. Specifically:
> - In **position-aware customization**, performance degrades when the user-provided mask is **extremely ambiguous or excessively small**.
> - In **position-free customization**, performance can also decrease when the reference identity lacks sufficient detail or when the texture is too complex.
>
> We have included a detailed analysis of these scenarios in the newly added **Failure Case** section (**Sec. I, Fig. 14**) of the Appendix.

---

### Official Review · Reviewer_YcXu · 2025-11-01

**Soundness:** 3
**Presentation:** 3
**Contribution:** 3
**Rating:** 6
**Confidence:** 2

**Summary:**

This paper proposes IC-Custom, a unified framework for both position-aware (e.g., image insertion, try-on) and position-free (e.g., text-guided generation) image customization. It reformulates both tasks through a diptych in-context formulation, processed by a new In-Context Multi-Modal Attention (ICMA) module with task-register tokens and boundary-aware positional embeddings. A related dataset (CustomData) and a new benchmark are also introduced. IC-Custom achieves strong SOTA results while fine-tuning only 0.4 % of the base model.

**Strengths:**

+ Elegant unification: A clear and practical integration of position-aware and position-free customization within one architecture, avoiding the need for separate specialized models.

+ Well-designed ICMA mechanism: Task tokens and boundary embeddings effectively address task ambiguity and spatial confusion, with strong ablations supporting their contribution.

+ Comprehensive evaluation: Extensive comparisons show consistent superiority over GPT-4o and recent open-source baselines, supported by high-quality qualitative examples.

**Weaknesses:**

+ The “In-context learning” framing is somewhat overstated—the model is trained end-to-end rather than showing adaptive few-shot behavior.

+ There is no explicit modeling of 3-D or geometric consistency; limited robustness tests for large viewpoint changes.

**Questions:**

1. How do task tokens learn to specialize—via supervision or implicitly from data mixing?

2. Can IC-Custom handle multi-reference or style-blending scenarios?

---

> ### Author Response · Authors · 2025-11-20
>
> We appreciate the reviewer’s thoughtful comment.
>
> > **Q1: [Weakness 1]** The “In-context learning” framing is somewhat overstated—the model is trained end-to-end rather than showing adaptive few-shot behavior.
>
> **A1:**
>
> **[Clarification on “In-context” Terminology]:**
> We fully understand the reviewer’s concern. In NLP, *In-context learning (ICL)* traditionally refers to **few-shot reasoning based on explicit demonstration examples**, where a model observes several (e.g., {problem, solution}) demonstrations and infers patterns at test time without parameter updates.
>
> However, in the **image generation and editing community**, the notion of “in-context” has evolved and diverges from the NLP definition. Recent works such as **IC-LoRA[1]** and **IC-Edit[2]** use “in-context” to denote operations such as:
> - concatenating reference images rather than tokens,
> - jointly feeding the captions of both the reference image and the target image.
>
> Within this convention, “in-context” denotes joint conditioning on the reference image and the target caption, instead of demonstration-based few-shot reasoning as in NLP. Our terminology is consistent with prior works in generative models.
>
>
> **[Revision]:**
> To avoid ambiguity, we have added a clarifying footnote at the first occurrence of “In-context” in **Sec. 3.1 (p. 5)**, explicitly distinguishing NLP-style ICL from the usage adopted in generative methods. We hope this clarification addresses the reviewer’s concern.
>
>
> [1] Huang, L., Wang, W., Wu, Z.F., Shi, Y., Dou, H., Liang, C., Feng, Y., Liu, Y. and Zhou, J., 2024. In-context lora for diffusion transformers. arXiv preprint arXiv:2410.23775.
> [2] Zhang, Z., Xie, J., Lu, Y., Yang, Z. and Yang, Y., 2025. In-context edit: Enabling instructional image editing with in-context generation in large scale diffusion transformer. arXiv preprint arXiv:2504.20690.
>
>
> > **Q2: [Weakness 2]** There is no explicit modeling of 3-D or geometric consistency; limited robustness tests for large viewpoint changes.
>
> **A2:**
>
> ### 1. Validation on Geometric Consistency
>
> **[Methodology]:**
> To verify whether our customization preserves the underlying geometric structure of the reference identity, we conducted a validation experiment using **VGGT [1]** to assess the geometric alignment between the customized results and the reference inputs.
>
> **[Findings]:**
> The results indicate that our method implicitly captures the latent geometric structure and maintains robust alignment with the reference subject. We attribute this to the inclusion of **real-world data** in our training, enabling the model to implicitly learn robust 3D priors.
>
> **[Revision]:**
> The results are now included in **Appendix Sec. E (Fig. 11)**.
>
> [1] Wang, Jianyuan, et al. "Vggt: Visual geometry grounded transformer." Proceedings of the Computer Vision and Pattern Recognition Conference. 2025.
>
> ### 2. Robustness to Large Viewpoint Changes
>
> **[Evidence of Robustness]:**
> Regarding the concern about limited robustness tests, we respectfully point out that examples of large viewpoint changes were included in the appendix of our original submission. In the revised manuscript, these are presented in **Appendix Fig. 17** and **Fig. 18**, which specifically include cases exhibiting viewpoint variations:
> * **The Sloth Plushie (Fig. 17, 2nd row from bottom)**;
> * **The Cat (Fig. 17, 2nd row from bottom)**;
> * **The Lamp (Fig. 18, 2rd row)**.
> * **The Rabbit (Fig. 18, 3rd row)**.
>
> Both examples exhibit significant viewpoint variations while retaining high identity fidelity.
>
> **[Clarification]:**
> We acknowledge that our framework prioritizes **identity preservation and compositional harmony** over *explicit pose control*. Such fine-grained pose controllability falls under a different research branch and is beyond the scope of our current work, though it represents a promising direction for future exploration.

---

> > ### Author Response · Authors · 2025-11-20
> >
> > > **Q3: [Questions 1]** How do task tokens learn to specialize—via supervision or implicitly from data mixing?
> >
> > **A3:**
> >
> > **[Mechanism]:**
> > The **Task-oriented Register tokens (TR)** are learned **implicitly** in an end-to-end manner, driven by the mixture of task-specific training data rather than explicit auxiliary supervision.
> >
> > **[Unified Customization Tasks]:**
> > Specifically, our framework unifies three distinct customization scenarios:
> > 1.  **Position-free customization**;
> > 2.  **Position-aware customization** (conditioned on precise masks);
> > 3.  **Position-aware customization** (conditioned on coarse, user-drawn masks).
> >
> > **[Implementation Details]:**
> > We initialize the TR as learnable parameters $\in \mathbb{R}^{3 \times N \times D}$ (where $3$ represents the task types, $N$ the number of tokens, and $D$ the dimension). During training, distinct TR embeddings are added to the input sequence based on the data variant corresponding to the specific task. This design effectively enables the model to distinguish between scenarios and enhances performance across all three tasks.
> >
> >
> > > **Q4: [Questions 2]** Can IC-Custom handle multi-reference or style-blending scenarios?
> >
> > **A4:**
> >
> > ### 1. Multi-Reference Customization
> >
> > **[Seamless Scalability]:**
> > In the original submission, we mentioned that our method can be extended to **multi-reference customization**, enabling finer-grained identity consistency. Preliminary results were provided in the appendix.
> >
> > **[Revision & Evidence]:**
> > In the revised manuscript, we have polished and expanded this section, adding **more detailed explanations** as well as the **mathematical formulation** of the multi-reference extension. The updated content has been moved to **Appendix Sec.F and Fig.12**, demonstrating that our approach—supported by the proposed **ICMA module**—extends naturally to **multi-reference** scenarios.
> >
> > **[Clarification]:**
> > Multi-reference customization is not part of the core claim of our paper; rather, it represents a **scalable extension** of our framework. We include these preliminary results to illustrate the model's potential and to inspire future exploration in this direction.
> >
> > ### 2. Style-Blending Scenarios
> >
> > **[Cross-Domain Capabilities]:**
> > While we included some cross-domain customization examples in the original submission, we have further strengthened this aspect to address the reviewer's comment.
> >
> > **[New Validation]:**
> > To provide clarity, we have added **Appendix Sec.G and Fig.13** to demonstrate that our method effectively supports cross-domain customization tasks. This section explicitly verifies our method's capability in handling cross-domain tasks, including **style-blending scenarios**, demonstrating robust performance beyond realistic scenarios.
> >
> > **[Clarification]:**
> > However, we clarify that explicit style-content disentanglement is not a primary objective of our work. As such, it is challenging to specify a reference style image as input to achieve more specific style-driven customization.

---

### Official Review · Reviewer_uenq · 2025-11-01

**Soundness:** 3
**Presentation:** 3
**Contribution:** 3
**Rating:** 6
**Confidence:** 4

**Summary:**

The authors propose IC-Custom, a unified framework integrating position-aware (mask-guided editing) and position-free (text-guided identity generation) image customization. It adopts an in-context diptych format by concatenating reference and fill-in images into a unified input and introduces ICMA module, which incorporates task-oriented learnable register tokens and boundary-aware embeddings to resolve task ambiguity and boundary confusion. They also curate CustomData (12K samples: 8K real-world e-commerce diptychs, 4K filtered synthetic ones) to avoid unrealistic synthetic data flaws, with strict filtering (DINOv2 similarity ≥0.2) and auto-annotation (Qwen-VL2.5 for captions, Grounded SAM for masks). Experiments results show the effectiveness of the proposed method.

**Strengths:**

1. The paper addresses the long-standing gap of isolated position-aware and position-free image customization by proposing a unified framework.
2. The method balances performance and efficiency: built on pre-trained FLUX.1-Fill, it only trains 0.4% of parameters via LoRA fine-tuning, while outperforming baselines on ProductBench and DreamBench.
3. Overall, the writing is clear.

**Weaknesses:**

1. Limited Scenario Generalization: IC-Custom only validates single-object customization. Its performance in multi-object scenarios (e.g., inserting a backpack and a laptop simultaneously) remains untested.
2. The paper does not mention any performance in the absence of text input. Position-aware customization relies on text to define identity and scene constraints. How the model behaves when it lacks clear text guidance is unclear.

**Questions:**

How the model behaves when it lacks clear text guidance?

---

> ### Author Response · Authors · 2025-11-20
>
> We appreciate the reviewer’s insightful comment.
>
>
> > **Q1: [Weakness 1]** Limited Scenario Generalization: IC-Custom only validates single-object customization. Its performance in multi-object scenarios (e.g., inserting a backpack and a laptop simultaneously) remains untested.
>
> **A1:**
>
> **[Clarification]:**
> IC-Custom is designed for **unified, feed-forward and single-object customization**. Multi-object customization is an **extension** of our setting and is not claimed as a primary contribution.
>
> **[Compositionality]:**
> Importantly, IC-Custom is **naturally composable**. Multi-object customization can be achieved via **multi-round iterative operation** (e.g., insert backpack → feed output back → insert laptop). This iterative workflow is consistent with common multi-step interactive editing practices (e.g., gpt–style customization), offering flexible and user-friendly control.
>
> **[Revision]:**
> We have added **multi-round multi-object qualitative results** in the revised manuscript (**Appendix Sec.D, Fig.10**). We kindly refer the reviewer to the updated section. Results demonstrate that IC-Custom can reliably perform multi-object customization in a sequential manner without requiring any architectural changes or additional training.
>
>
> > **Q2: [Weakness 2]&[Question 1]** Position-aware customization relies on text to define identity and scene constraints. How the model behaves when it lacks clear text guidance is unclear.
>
> **A2:**
>
> **[Clarification]:**
> In our position-aware customization setting, **no-text input is fully supported**, rather than being an unsupported corner case. IC-Custom can operate under four conditioning modes:
>
> (i) **Full-conditions** (default),
> (ii) **Redux-only** (no text),
> (iii) **Text-only** (no Redux reference),
> (iv) **Neither text nor Redux**, relying solely on the **in-context diptych reference image**.
>
> **[Behavior of Distinct Input Modes]:**
> We conducted an ablation study on ProductBench to evaluate all four configurations. As shown below, IC-Custom maintains stable and reliable position-aware customization across all input-conditioning settings:
>
> | Setting                             | Precise Mask DINO-I ↑ | Precise Mask CLIP-I ↑ | Precise Mask CLIP-T ↑ | User-drawn Mask DINO-I ↑ | User-drawn Mask CLIP-I ↑ | User-drawn Mask CLIP-T ↑ |
> |-------------------------------------|------------------------|------------------------|------------------------|---------------------------|---------------------------|---------------------------|
> | **Full conditions (default)** | *63.14*                 | **81.92**                 | **31.75**                 | *63.28*                   | **81.95**                    | **31.80**                    |
> | Redux-only               | **63.34**                 | *81.49*                | *31.37*                | **63.78**                    | *81.86*                    | 31.10                    |
> | Text-only                | 62.96                 | 81.32                 | 31.42                 | 63.04                    | 81.43                    | *31.29*                    |
> | Diptych-only              | 62.97                 | 81.41                 | 31.41                 | 63.18                    | 81.77                    | 31.30                   |
>
> This robustness arises from our training-time random dropping of conditioning signals, which enables the model to adapt reliably to diverse input configurations. Notably, the diptych-only mode already performs well; the Redux Encoder further enhances identity preservation, the Text Encoder complements image cues with textual semantics, and the full-conditions setting provides the most stable and robust results. The qualitative visualizations further corroborate these observations.
>
> **[Revision]:**
> We have included the full **quantitative and qualitative ablations** for all four input-conditioning modes in the revised manuscript (**Appendix Sec.B, Tab.4, Fig.8**). The results show that IC-Custom performs strongly across all input modes: the full-conditions setting is the most stable overall, while the reduced-input variants also deliver consistently competitive results.

---

### Author Response · Authors · 2025-11-20

We sincerely thank all reviewers for the time and effort spent evaluating our submission and for the constructive feedback provided.

We have responded to all comments in detail and revised the manuscript accordingly. The updated PDF includes additional analyses, clarifications, and new experimental results, all highlighted in red for ease of reference. The main revisions appear on **pp. 9–10** of the main text and in the **Appendix (pp. 15–20)**. We kindly invite the reviewers to examine the revised content, and we would be very happy to further discuss any remaining questions or suggestions.

---

### Author Response · Authors · 2025-12-03
**Response Summary**

We sincerely thank the Area Chairs and all reviewers for their time and insightful feedback.

Reviewers consistently acknowledged the elegance of our **unified paradigm** (uenq, YcXu, JiRo, pHSt), the **novelty** of our approach (uenq, YcXu, JiRo, pHSt), the **thoroughness of our evaluation** (YcXu, JiRo), and the **clarity of the presentation** (uenq, pHSt). We are deeply encouraged by this consensus, as well as by the generally positive initial scores.

In the revision and rebuttal, we carefully addressed all concerns. The key improvements are summarized below:

---

## 1. Conceptual Clarification
**(Raised by YcXu)**

### Issue
- Ambiguity around the meaning of “in-context learning” in generative modeling.
- Whether Task Register tokens (TR) are explicitly supervised or implicitly learned.

### Resolution
- Added a footnote clarifying the **generative-model interpretation of “in-context”**, consistent with IC-LoRA and IC-Edit.
- Explained that TR are **implicitly learned end-to-end**, driven by task-diverse training data rather than auxiliary supervision, with additional details added in the response.

---

## 2. Geometry Preservation & Challenging Scenarios
**(Raised by uenq, YcXu, JiRo)**

### Issue
- Need for stronger evidence on multi-object, multi-reference, geometric consistency, large viewpoint changes, complex scenes, and failure cases.

### Resolution
- Added **multi-round multi-object** customization (App. Sec.D, Fig.10).
- Added **multi-reference** customization (App. Sec.F, Fig.12).
- Added **VGGT-based geometric alignment validation** (App. Sec.E, Fig.11).
- Included examples with **large viewpoint variation** (App. Fig.17–18).
- Added **complex scenes** and **cross-domain** customization scenarios (App. Sec.G, Fig.13, Fig.18).
- Added **failure-case analysis** (App. Sec.I, Fig.14).

---

## 3. Effectiveness of Model & Input Components
**(Raised by uenq, pHSt)**

### Issue
- Whether the method remains effective without text input.
- Necessity of the proposed Task-oriented Register tokens (TR) and Boundary-aware Positional Embeddings (PE).
- Whether identity consistency is overly dependent on the Redux Encoder.

### Resolution
- Added comprehensive ablations across **four conditioning modes** (full / Redux-only / text-only / diptych-only), demonstrating that the model remains **highly robust even without text** (App. Sec.B, Tab.4, Fig.8).
- Moved and refined **qualitative ablations for TR/PE** into the main paper (Sec.4.5, Fig.6–7). Results show that removing TR/PE leads to task ambiguity and boundary artifacts, confirming their necessity.
- Clarified the role of the Redux Encoder: the **FLUX.1 workflow (FLUX.1-Fill + FLUX.1-Redux)** already serves as a Redux-based baseline, and our method significantly outperforms it. In addition, we include a new **Redux-free ablation**, showing that the Redux Encoder is optional and that **IC-Custom works strongly even without it** (App. Sec.B, Tab.4, Fig.8).

---

## 4. Dataset Diversity, Training Cost & Inference Efficiency
**(Raised by JiRo)**

### Issue
- Need clearer description of dataset composition, compute requirements, and inference-time competitiveness.

### Resolution
- Expanded dataset details: real-world product data + SynCD (75K Objaverse rigid assets + 16 deformable super-categories with ~100 subspecies).
- Added full compute profile: **8×H20 GPUs**, ~160 GPU-hours for 20K training iterations.
- Added inference-time comparison (~30s), on par with DreamO, OmniCtrl, and InsertAnything under identical settings.

---


We have marked all revisions in the PDF using $\color{red}red$ $\color{red}text$, including updates in Sec.4.5 and Appendix Sec.B, Sec.D, Sec.E, Sec.F, Sec.G, and Sec.I. We hope these substantial revisions and clarifications fully address the reviewers’ concerns and further strengthen the paper for the final decision.

---

### Meta-Review · Area_Chair_trrB · 2026-01-12

**Summary:**

This paper introduces IC-Custom, a unified framework for position-aware and position-free customized generation tasks. The reviewers raises concerns on the following aspects: 1. model's geometry preservation capability under challenging scenarios like large pose change and complex background. 2. Thorough ablation on model components. 3. computation cost. The authors addressed most of them by adding more experiment results and clearer explanations. After the rebuttal, all the reviewers give a score of 6. Based on these, I recommend accepting this paper.

**Reviewer Concerns:**

The rebuttal and revision addressed the reviewers' concerns on model's geometry preservation capability and it performance on challenging scenarios, the effectiveness of model components, and the clarification on computation cost. Overall, most concerns from the reviewers have been addressed.

**Reviewer Scores:**

The reviewers have provided detailed comments in the first round, but did not have follow up discussions.

---

### Decision · Program_Chairs · 2026-01-26

Accept (Poster)